# Unveiling IL6R and MYC as Targeting Biomarkers in Imatinib-Resistant Chronic Myeloid Leukemia through Advanced Non-Invasive Apoptosis Detection Sensor Version 2 Detection

**DOI:** 10.3390/cells13070616

**Published:** 2024-04-02

**Authors:** Chia-Hwa Lee, Kai-Wen Hsu, Yao-Yu Hsieh, Wei-Ting Li, Yuqing Long, Chun-Yu Lin, Shu-Huey Chen

**Affiliations:** 1School of Medical Laboratory Science and Biotechnology, College of Medical Science and Technology, Taipei Medical University, New Taipei City 23561, Taiwan; chlee@tmu.edu.tw; 2Ph.D. Program in Medicine Biotechnology, College of Medical Science and Technology, Taipei Medical University, New Taipei City 23561, Taiwan; 3TMU Research Center of Cancer Translational Medicine, Taipei Medical University, Taipei 11031, Taiwan; 4Center for Intelligent Drug Systems and Smart Bio-Devices (IDS2B), National Yang Ming Chiao Tung University, Hsinchu 30068, Taiwan; chunyulin@nycu.edu.tw; 5Research Center for Cancer Biology, China Medical University, Taichung City 40402, Taiwan; kwhsu@mail.cmu.edu.tw; 6Institute of Translational Medicine and New Drug Development, China Medical University, Taichung City 40402, Taiwan; 7Program for Cancer Biology and Drug Discovery, Drug Development Center, China Medical University, Taichung City 40402, Taiwan; 8Division of Hematology and Oncology, Shuang Ho Hospital, Taipei Medical University, New Taipei City 23561, Taiwan; 10573@s.tmu.edu.tw; 9Division of Hematology and Oncology, Department of Internal Medicine, School of Medicine, College of Medicine, Taipei Medical University, Taipei 11031, Taiwan; 10Department of Physiology, UT Southwestern Medical Center, Dallas, TX 75390, USA; liwatin@gmail.com; 11Nuffield Department of Medicine, University of Oxford, Oxford OX3 7BN, UK; yuqing.long@ndm.ox.ac.uk; 12Chinese Academy of Medical Science Oxford Institute, University of Oxford, Oxford OX3 7BN, UK; 13Institute of Bioinformatics and Systems Biology, National Yang Ming Chiao Tung University, Hsinchu 30068, Taiwan; 14School of Dentistry, Kaohsiung Medical University, Kaohsiung 807378, Taiwan; 15Department of Pediatrics, School of Medicine, College of Medicine, Taipei Medical University, Taipei 11031, Taiwan; 16Department of Pediatrics, Shuang Ho Hospital, Taipei Medical University, New Taipei City 23561, Taiwan

**Keywords:** apoptosis, NIADS, Imatinib-resistant, Chronic Myeloid Leukemia, CRISPR/Cas9, gene editing

## Abstract

The management of chronic myelogenous leukemia (CML) has seen significant progress with the introduction of tyrosine kinase inhibitors (TKIs), particularly Imatinib. However, a notable proportion of CML patients develop resistance to Imatinib, often due to the persistence of leukemia stem cells and resistance mechanisms independent of BCR::ABL1 This study investigates the roles of IL6R, IL7R, and MYC in Imatinib resistance by employing CRISPR/Cas9 for gene editing and the Non-Invasive Apoptosis Detection Sensor version 2 (NIADS v2) for apoptosis assessment. The results indicate that Imatinib-resistant K562 cells (K562-IR) predominantly express IL6R, IL7R, and MYC, with IL6R and MYC playing crucial roles in cell survival and sensitivity to Imatinib. Conversely, IL7R does not significantly impact cytotoxicity, either alone or in combination with Imatinib. Further genetic editing experiments confirm the protective functions of IL6R and MYC in K562-IR cells, suggesting their potential as therapeutic targets for overcoming Imatinib resistance in CML. This study contributes to understanding the mechanisms of Imatinib resistance in CML, proposing IL6R and MYC as pivotal targets for therapeutic strategies. Moreover, the utilization of NIADS v2 enhances our capability to analyze apoptosis and drug responses, contributing to a deeper understanding of CML pathogenesis and treatment options.

## 1. Introduction

Cancer has emerged as one of the most formidable diseases affecting human health globally in recent decades [1]. Apoptosis, a natural defense mechanism against the uncontrolled growth of cancerous cells, plays a crucial role in anti-cancer strategies [2]. The dysregulation of apoptotic pathways poses significant challenges in clinical cancer treatment, particularly in terms of cancer cell proliferation and resistance to chemotherapy. This pathological resistance underscores the critical need for precise, reliable, and sensitive methodologies capable of detecting apoptosis with high fidelity. While conventional techniques, such as cell morphology observation under microscopy, DNA fragmentation assays, and Western blot, have provided valuable insights into apoptotic processes, they have limitations, such as labor-intensive procedures, large sample requirements, and limited sensitivity to subtle changes in apoptosis that influence treatment outcomes [3]. Furthermore, the inability of these traditional methods to support real-time or high-throughput analysis has become increasingly evident, indicating a gap in current research tools that requires attention to advance cancer therapy and research.

Chronic Myeloid Leukemia (CML) is a type of blood cancer characterized by the presence of the BCR::ABL1 fusion gene, commonly known as the Philadelphia chromosome [4]. The introduction of tyrosine kinase inhibitors (TKIs), with Imatinib being the initial first-line treatment, has significantly revolutionized the management of CML. However, the emergence of Imatinib resistance in approximately one-third of treated patients signals a critical barrier to sustained treatment efficacy [5], leading to disease progression and necessitating the exploration of alternative therapeutic strategies [6,7]. The resistance to Imatinib is multifaceted, involving the occurrence of adverse reactions to the drug in some patients and the emergence of over 1000 substitution mutations within the BCR::ABL1 kinase domain. These mutations, particularly those that directly impede Imatinib binding [4,8], such as those in critical regions like the nucleotide-binding (P-loop) and activation loop (A-loop) [9], diminish the drug’s efficacy, highlighting the intricate interplay between the drug and its target.

In response to this complex issue, the treatment options for CML have expanded to include second-line TKIs, such as Dasatinib [10], Nilotinib [11], and Bosutinib, each with distinct mechanisms of action against various BCR::ABL1 mutations. Ponatinib, a third-generation TKI, specifically targets the challenging T315I mutation, which confers high resistance to CML cells by altering the ATP-binding domain of BCR::ABL1 [12]. This mutation hinders TKI access while allowing ATP to continue fueling cancerous activity. The presence of the T315I mutation not only exemplifies the adaptive resistance mechanisms in CML but also emphasizes the critical need for a personalized therapeutic approach based on the genetic characteristics of the disease in individual patients [13]. The clinical implications of the T315I mutation extend beyond its mechanistic complexity, as it is associated with significantly poorer survival rates, underscoring the urgent requirement for ongoing research and the development of more effective, mutation-specific treatment strategies. By deepening the understanding of the genetic basis of TKI resistance and utilizing the latest advancements in targeted therapy, there is a promising avenue for enhancing the prognosis and quality of life for individuals grappling with CML [14].

Recent advancements in CML treatment have brought to light the limitations of strategies focused solely on BCR::ABL1-dependent mechanisms in patients with CML resistant to TKIs [15]. New evidence emphasizes the crucial role of BCR::ABL1-independent pathways, particularly the importance of leukemia stem cells that persist and sustain the disease even in the absence of BCR::ABL1 kinase activation [16]. These stem cells, which share molecular components with pluripotent hematopoietic stem cells but exhibit distinct biological properties, present challenges as well as opportunities for targeted therapies [17]. A detailed understanding of the survival mechanisms of these cells and their self-renewal capacity opens up possibilities for treatment strategies aimed at addressing the root cause of TKI resistance [18]. Furthermore, the varying responses to second-line therapies in the context of Imatinib resistance necessitate a comprehensive evaluation of their efficacy, highlighting the importance of identifying specific biomolecular markers associated with drug resistance. These markers could shed light on pathways to enhance cell viability and drug responsiveness, thereby refining therapeutic approaches. This holistic approach not only aims to overcome immediate challenges related to drug resistance but also sets the stage for future research that could potentially eradicate CML at its core, leading to significant improvements in patient outcomes.

Robust detection of apoptosis is crucial for understanding cancer biology and assessing the efficacy of cancer treatments [19]. In the context of personalized medicine, there is an increasing demand for rapid and precise apoptosis measurement techniques. Such methods are essential for monitoring treatment effectiveness and informing therapeutic decisions to enhance patient outcomes. To meet this need, our team has developed the Non-Invasive Apoptosis Detection Sensor (NIADS). This bioluminescence-based assay represents a transformative approach to detecting and studying apoptosis [20]. The NIADS technology is distinguished by its rapid, non-invasive, and highly sensitive detection of apoptotic events, thus addressing the drawbacks inherent in traditional methods. Integrating the analytical rigor of molecular biology with the practicality of luminescence imaging, this novel sensor facilitates real-time observation of apoptosis, thereby revealing the dynamic nature of cell death processes. Furthermore, this study employs the advanced NIADS v2 sensor, featuring improved functionalities for quantifying cell populations and normalizing caspase-3 activity via integrated reporters. This enhancement provides a reliable measure of cellular apoptotic events. By integrating CRISPR/Cas9 gene editing with the advanced capabilities of the NIADS v2 sensor, this investigation aims to elucidate the complex roles of specific genes in Imatinib resistance and assess their viability as targets for novel therapeutic strategies.

## 2. Materials and Methods

### 2.1. Cell Culture and Transfection

The human CML cell line (K562) was generously provided by Dr. Kai-Wen Hsu from the Research Center for Cancer Biology at China Medical University, Taichung, Taiwan. The K562-IR cells, a derivative of the K562 cell line, were developed through a two-month exposure to 0.05 μM of Imatinib (Sigma-Aldrich, St. Louis, MO, USA), followed by one month of exposure to increasing Imatinib concentrations of 0.1, 0.5, 1, and 5 μM [21]. During the K562-IR cell line establishment, the culture medium was refreshed weekly. The cells were cultured in Dulbecco’s Modified Eagle’s Medium: Nutrient Mixture F-12 (Gibco; Thermo Fisher Scientific, Inc., Waltham, MA, USA), and supplemented with 10% (*v*/*v*) fetal bovine serum (Biological. Industries, Kibbutz Beit Haemek, Israel), along with 100 units/mL of penicillin and 100 mg/mL of streptomycin (Gibco; Thermo Fisher Scientific, Inc., Waltham, MA, USA). Cell cultures were maintained at 37 °C in an incubator with 5.0% CO_2_. Lentiviral transfection was conducted using a multiplicity of infection (MOI) of 5 for 72 h, followed by antibiotic selection with either neomycin (Sigma-Aldrich, St. Louis, MO, USA) or puromycin (Sigma-Aldrich, St. Louis, MO, USA) for 48 h. The genomic Indel (insertions or deletions) profiling was assessed using Sanger sequencing and immunoblotting.

### 2.2. Production of Lentiviral Particles for IL6R, IL7R, and MYC KO

Lentiviral particles targeting *IL6R*, *IL7R*, *MYC*, and NIADS v2 were generated through a transient transfection process involving Phoenix-ECO cells (ATCC, Manassas, VA, USA). This process was facilitated using TransIT^®^-LT1 Reagent from Mirus Bio LLC in Madison, WI, USA. Guide oligonucleotides were subjected to phosphorylation and annealing and then cloned into the lentiCRISPR v2 vector (Addgene, Watertown, MA, USA, #52961). Sequencing was performed to confirm the integrity of all plasmid constructs. To produce lentiviral particles, plasmids were co-transfected with pMD2.G (Addgene, Watertown, MA, USA, #12259) and psPAX2 (Addgene, Watertown, MA, USA, #12260), both graciously provided by Didier Trono at EPFL, Lausanne, Switzerland. Lentiviral particles were harvested at 36 and 72 h post-transfection and subsequently concentrated using Lenti-X Concentrator^®^ (Clontech, Mountain View, CA, USA). The concentration of lentivirus for each gene was quantified utilizing Q-PCR, thus ensuring accurate dosing and delivery of these genetic materials to the target cells.

### 2.3. Sanger Sequencing and Gene Editing Efficiency Assay

Genomic DNA was extracted, and the target genes were PCR-amplified using the following primers: IL6R, forward TCAGAGTGGCAGGGAAG and reverse CACCACACCCAGCTAAT; IL7R, forward TAAATCATCACTAAGTATCATAGCAGC and reverse GCTCCAGTTAGCCACTTC; and MYC, forward AAGGGTGCTCCCTTTATT and reverse GAGGCCAGCTTCTCTGA. The PCR products were purified using a PCR Clean-up Purification Kit (Geneaid Biotec, New Taipei City, Taiwan) and sequenced through Sanger sequencing using the forward PCR primers. The editing efficiency of the sgRNAs and the potential induced mutations were assessed using TIDE v3.3.0 software (https://tide-calculator.nki.nl; accessed on 18 January 2024; Netherlands Cancer Institute), which required only two Sanger sequencing runs from wild-type cells and mutated cells.

### 2.4. Real-Time Quantitative Polymerase Chain Reaction (Q-PCR)

Primers for the ALDH1 region (forward 5′-GAGTGTTGAGCGGGCTAA-3′ and reverse 5′-CTCCTCCACATTCCAGTTTG-3′), CD44 region (forward 5′-GTCGAAGAAGGTGTGGG-3′ and reverse 5′-GGTCTGGAGTTTCTGACG-3′), CD47 region (forward 5′-CTCCTTCGTCATTGCCATA-3′ and reverse 5′-AACTAGTCCAAGTAATTGTGCT-3′), and β-glucuronidase GUS region (forward 5′-CTCCTTCGTCATTGCCATA-3′ and reverse 5′-CCTTTAGTGTTCCCTGCTAGAATA-3′) were used for gene quantification. All oligo primers were synthesized by Genomics BioSci and Tech (Taipei, Taiwan). A LightCycler thermocycler (Roche Molecular Biochemicals, Mannheim, Germany) was used for Q-PCR analysis. One microliter of sample and master mix (Roche, Basel, Switzerland) was first denatured for 10 min at 95 °C and then subjected to 40 cycles (denaturation at 95 °C for 5 s; annealing at 60 °C for 5 s; and elongation at 72 °C for 10 s) with detection of fluorescence intensity. Gene expression was finally normalized to GUS expression using the built-in Roche LightCycler Software, version 4 [21].

### 2.5. RNA Library Preparation, Sequencing, and Analysis

RNA sequencing was performed by Taiwan Genomic Industry Alliance lnc. (New Taipei City, Taiwan). RNA from K562 and K562IR was quantified using Bioanalyzer 2100 (Agilent, Santa Clara, CA, USA) with RNA Nano Chip. The RNA sequence (RNASeq) procedures were carried out according to the manufacture’s protocol from Illumina. Library construction of samples was conducted using the Truseq stranded mRNA kit (Illumina, San Diego, CA, USA) for 150 bp paired-end sequencing on Novaseq 6000. The sequencing was determined using sequencing-by-synthesis technology via NovaSeq 6000 S4 Reagent Kit v1.5 (300 cycles). Raw sequences were obtained from the Illumina Pipeline Software (bcl2fastq v2.20.0.422) and expected to generate 20 M (million reads) per sample. The quality of the raw reads was checked using FastQC (v0.11.9). Adapters and low-quality bases and reads were trimmed with Trimmomatic (v0.36) using the parameters ILLUMINACLIP:Adapter.fa:2:30:10 LEADING:3 TRAILING:3 SLIDINGWINDOW:4:15 MINLEN:36. After trimming, the clean reads were aligned to the reference genome using HISAT2 (v2.2.1). StringTie (2.2.1) was used to perform read count normalization. FeatureCounts (v1.6.2) was used to perform read summarization. Differential expression (DEG) analysis was performed using DEGseq (v2.2.1) without biological replicates or DESeq^2^ (1.34.0) with biological replicates. Significant DEG sets were filtered by the absolute value of log^2^fold change ≥ 2 and adjusted *p*-value < 0.005 according to the results of DEGseq. To identify biological processes and pathways that are significantly enriched by the differentially expressed genes, KEGG (Kyoto Encyclopedia of Genes and Genomes) and GO (Gene Ontology) were used for annotation.

### 2.6. MTT Viability Assay

Cell viability was assessed using the 3-(4,5-dimethylthiazol-2-yl)-2,5-diphenyltetrazolium MTT assay (Sigma, M5655), a method relying on the reduction of yellow MTT to purple formazan by viable cells [22,23]. In 96-well plates, 1 × 10^5^ K562 or K562-IR cells were seeded and allowed to adhere overnight before exposure to various concentrations of dimethyl sulfoxide (DMSO) or anti-cancer agents, following the experimental protocol. After 48 h of treatment, the culture medium was replaced with fresh medium containing 1 μg/mL of MTT. Following a two-hour incubation period, 100 μL of DMSO (Sigma, D8418) was added to each well, and the absorbance at 570 and 630 nm was determined. The percentage of cell viability was calculated using the formula:Percentage viability = (OD sample − OD medium)/(OD control − OD medium) × 100%

This assay provided a quantitative measure of cell viability and was instrumental in assessing the effects of different treatments on K562 and K562-IR cells.

### 2.7. Bioluminescence Assay through IVIS

Bioluminescence imaging was conducted using a highly sensitive, cooled charge-coupled device (CCD) camera, housed within a light-tight specimen box (In Vivo Imaging System-IVIS; Xenogen, Hopkinton, MA, USA). K562 cells were seeded and exposed to various concentrations of dimethyl sulfoxide (DMSO) or anti-cancer agents for 8 h. A multiple-well plate containing the treated cells was then subjected to 1.5 mg/mL of D-luciferin (Promega, Madison, WI, USA) and positioned on a heated stage inside the camera box during the imaging process. The emitted light from cells was captured by the IVIS camera system and subsequently integrated, digitized, and displayed for analysis. Regions of interest within the displayed images were identified, and the total photon count was quantified utilizing Living Image^®^ software 4.0 (Caliper, Alameda, CA, USA).

### 2.8. Protein Extraction, Western Blot, and Antibodies

For Western blot analysis, both K562 and in vitro K562-IR cells were harvested and subjected to a single wash with ice-cold, phosphate-buffered saline (PBS). Subsequently, radioimmunoprecipitation assay (RIPA) lysis buffer (Thermo Scientific, Rockford, IL, USA), supplemented with protease inhibitors (Thermo Scientific, 78438), was added to the cells. A total of fifty micrograms of protein from each sample was resolved through sodium dodecyl sulfate polyacrylamide gel electrophoresis (SDS-PAGE) and subsequently transferred to a nitrocellulose membrane. Details regarding the primary antibodies and secondary antibodies employed in this study can be found in Appendix A. All primary antibodies were used at a 1:1000 dilution for overnight hybridization, followed by a one-hour incubation with secondary antibodies at a 1:4000 dilution. Band intensities were quantified using ImageJ v1.54g free software (Bethesda, MD, USA).

### 2.9. Flow Cytometry Analysis

To evaluate apoptosis and sub-G1 cell populations, K562-IR cells (1 × 10^6^ cells/dish) were seeded in 6 cm dishes and exposed to various concentrations of anti-cancer agents for 24 h. For sub-G1 apoptosis analysis, cells were harvested, washed with PBS, fixed in 75% alcohol, and labeled with Propidium Iodide (Sigma, P1304MP) to stain DNA. Additionally, apoptosis analysis was conducted using Annexin V-FITC/PI double staining. K562 cells (1 × 10^6^ cells/mL) were plated in a 6 cm dish and treated with TKIs for 24 h. Following harvesting, the cells were resuspended in 100 μL of binding buffer and incubated with Annexin V-FITC/PI solution in the dark for 15 min. Subsequently, the sub-G1 cell population and Annexin V-FITC/PI-labeled samples were analyzed using a flow cytometer (specifically, the FACS Verse (BD Biosciences, San Jose, CA, USA)).

### 2.10. Sphere Formation Assay

Similarly to the previous study [24], sphere formation was performed using K562 and K562-IR cells treated with various TKIs at different concentrations. The cells were cultured in serum-free DMEM/F12 medium supplemented with B27 (1:50), N2 (1:100), NEAA (1:100), 10 μg/mL of insulin, 0.5–20 μg/mL of hydrocortisol, 20 ng/mL of EGF, and 20 ng/mL of β-FGF for a duration of 18 days. All of the above supplements were supplied by Thermo Fisher Scientific Inc. (San Jose, CA, USA). Images of the formed spheres were captured under 200X magnification using a Zeiss Axio Observer Z1 inverted microscope (Carl Zeiss Microimaging GmbH, Jena, Germany), and the colony numbers were subsequently quantified.

### 2.11. Statistical Methods

The data generated in this study have been presented as the mean ± standard error (S.E.). All experiments were conducted at least twice, and the bioluminescent assays were conducted at least in triplicate. For analyzing pairwise samples, the Student’s *t*-test was employed. Statistical comparisons were conducted utilizing SigmaPlot (San Jose, CA, USA) and the Statistical Package for the Social Sciences v.13 (SPSS) from Chicago, IL, USA.

## 3. Results

### 3.1. The Advancements of Non-Invasive Apoptosis Detection Sensor v2 (NIADS v2)

In our previous study [20], we introduced the Non-Invasive Apoptosis Detection Sensor (NIADS), a platform designed for the rapid detection of apoptosis through quantification of caspase-3 activity in various cancer cells treated with histone deacetylation inhibitors (HDACi), including breast cancer, leukemia, and thyroid cancer cells [21,25]. In brief, a sensor with fusion luciferase fragments (Nluc and Cluc) carried peptide A (pepA) and peptide B (pepB) at the amino termini with 3X repeats of caspase-3 cleavage sequences (DEVD). Upon induction of apoptosis and caspase-3 activation, spontaneous cleavage at the DEVD site would free both pepA-Nluc and pepBCluc fragments and enable reconstitution of full-length luciferase through strong association of pepA and pepB peptides, resulting in bioluminescence activity from IADS with substrate addition. Despite NIADS’s significant advantages, certain limitations emerged, prompting the need for optimization to enhance its applicability in cancer research. One noted limitation was the observation that lower concentrations of cytotoxic drugs occasionally yield higher bioluminescent counts compared to higher concentrations. This unexpected result can be attributed to two possible reasons. Firstly, it is plausible that the drug may potentially interact with the luciferase protein, hindering its bioluminescent activity within the NIADS system. Secondly, highly cytotoxic drugs may directly induce cell death rather than initiating the caspase cascade required for the apoptotic events. To overcome these limitations and simultaneously preserve the rapid and precise apoptosis detection capabilities of NIADS, we modified the NIADS backbone by inserting a green fluorescent protein (GFP) reporter upstream of the NIADS sequence (Figure 1A) and ligating it with P2A and T2A autocleavage sequences (NIADS v2). This modification aimed to alleviate potential interaction challenges arising from the split-luciferase conformational change and activation by maintaining NIADS as a single molecule rather than a GFP-NIADS fusion protein.

To deliver NIADS v2 into K562 cells, we employed lentivirus transfection coupled with additional spin infection to enhance transfection efficiency in leukemia cells. Three days post-transfection, NIADS v2-K562 cells exhibited robust green fluorescence (Figure 1B). Subsequently, we employed Western blot to identify the different forms of bioluminescent proteins. The results clearly showed that the molecular weight of luciferase was approximately 65 kDa (Figure 1C), while NIADS protein had a molecular weight of 72 kDa. As anticipated, NIADS v2 protein contained both GFP and NIADS protein expressions, resulting in a higher molecular weight, which indicated successful plasmid construction and protein expression. In conclusion, NIADS v2 presents an enhanced platform for detecting cell apoptosis with improved precision and broader practicality for cancer research. The modified backbone and GFP reporter address the previous limitations of the earlier version, paving the way for future studies and applications.

### 3.2. Cytotoxicity Confirmation of RTKs Using Western Blot Analysis and NIADS v2

To evaluate the sensitivity of the NIADS v2 platform in detecting drug-induced apoptosis, NIADS v2-K562 cells were exposed to low (0.1 μM) and high (1 μM) concentrations of various anti-leukemia agents for 8 h (Figure 1D). Among the tested drugs, Dasatinib demonstrated the most significant apoptotic effects at both low and high concentrations. In contrast, high concentrations of Imatinib, Nilotinib, and Bosutinib induced moderate apoptotic effects, while their low concentrations did not show markedly stronger apoptotic signals compared to the background control. To ascertain the apoptosis detection threshold of NIADS v2, we monitored the bioluminescent activity in NIADS v2-K562 cells treated with varying concentrations (0.01–5 μM) of Imatinib, Dasatinib, Nilotinib, and Bosutinib (Figure 1E). With GFP normalization, the bioluminescent response indicated that higher concentrations of Dasatinib (>0.5 μM) achieved the maximum signal (approximately 2.5 times greater than the control). Even the minimal Dasatinib concentration (0.01 μM) exhibited a substantial apoptotic signal (1.4 times that of the control). In addition, Nilotinib was identified as the second most potent agent against K562 cells, causing a two-fold increase in bioluminescent activity with exposures to more than 1 μM of Nilotinib. Conversely, Imatinib and Bosutinib showed the least apoptotic impact on K562 cells, with about a 1.7-fold increase in bioluminescent activity at 5 μM of Imatinib and Bosutinib treatments, which gradually decreased with lower drug concentrations.

Subsequently, we performed a Western blot analysis to validate the apoptosis induction observed through the NIADS v2 assay. The results distinctly showed that ABL phosphorylation (p-ABL) was eliminated under all Dasatinib treatments (0.05–5 μM), while total ABL expression (T-ABL) remained high in K562 cells (Figure 1F). Moreover, aligning with prior findings, Dasatinib treatments notably triggered cell apoptosis, as indicated by the cleavage of NIADS, PARP, and caspase-3 proteins in NIADS v2-K562 cells, accompanied by a marked increase in the cell cycle regulatory proteins p21 and p27. Conversely, significant inhibition of ABL kinase activity and apoptotic signals, including NIADS, PARP, and caspase-3 protein cleavage, was observed only under high-concentration Imatinib treatment, with a minor increase in p27 protein levels (Figure 1G). Therefore, the NIADS v2 platform proves to be a reliable and consistent method for detecting apoptosis in the notoriously hard-to-transfect K562 leukemia cells. These results underscore the potential of this platform for assessing anti-leukemia agents and furthering cancer research [26].

### 3.3. Screening RTKs in Imatinib-Resistant K562 Cells Using NIADS v2

To address the challenge of Imatinib resistance in CML treatment, we utilized the NIADS v2 platform on Imatinib-resistant K562 cells (K562-IR) to explore potential therapeutic strategies and understand the regulatory mechanisms behind Imatinib resistance [27]. We evaluated the apoptotic responses of K562-IR cells to previously tested anti-K562 agents (Figure 2A). After GFP normalization, the data indicated that high concentrations of Dasatinib, Nilotinib, and Bosutinib significantly induced apoptosis in NIADS v2-K562-IR cells. Specifically, Dasatinib treatments above 0.5 μM generated considerable bioluminescent signals (exceeding 1.5-fold of the control), and significant apoptotic responses were also triggered by Nilotinib at 1 μM and 5 μM and Bosutinib at 5 μM (surpassing 1.5-fold of the control). Expectedly, NIADS v2-K562-IR cells showed no bioluminescent reaction to Imatinib treatment, underscoring K562-IR cells’ strong resistance to Imatinib. Subsequently, we utilized annexin V and PI staining to evaluate apoptotic populations under various TKI treatments. Flow cytometry analysis (Figure 2B) demonstrated that 5 μM of Imatinib significantly increased early apoptotic events in K562 cells (20.4 ± 3.2%, Figure 2C), whereas K562-IR cells exhibited a similar level of apoptosis (6.3 ± 2.1%) compared to the DMSO control (4.8 ± 1.6%). In contrast, treatment of both K562 and K562-IR cells with other TKIs showed considerable and similar levels of apoptosis after 24 h of drug exposure. This apoptosis assessment using NIADS v2 not only significantly reduces the duration required for drug exposure but also provides accurate and high-throughput measurements of apoptotic events. To corroborate the apoptosis detection outcomes garnered via NIADS v2, Western blot analyses were conducted. Following various Imatinib concentrations, K562 cells displayed pronounced expressions of PARP and caspase-3 cleavages (Figure 2D), while K562-IR cells exhibited negligible responses to Imatinib treatments (Figure 2E), highlighting the differential drug sensitivity between these cell lines.

In contrast, Dasatinib exhibited notable PARP and caspase-3 cleavage activities in both K562 and K562-IR cells, as evidenced in Figure 2F. Remarkably, K562-IR cells showed high sensitivity to Dasatinib treatment, even at the lowest concentration tested (0.1 μM), as depicted in Figure 2G. Beyond these findings, the blots also revealed that Dasatinib significantly reduced p-ABL kinase activity in both K562 and K562-IR cells, while Imatinib demonstrated more effective p-ABL kinase inhibition in K562 cells compared to K562-IR cells. These results imply that cells resistant to Imatinib might engage alternative molecular pathways or mechanisms to promote cancer progression. Dasatinib, on the other hand, seems to specifically target these altered molecules or pathways, leading to greater apoptotic effects than observed in Imatinib-sensitive leukemia cells. Identifying these crucial molecules or pathways could pave the way for developing new strategies aimed at enhancing the clinical outcomes of patients with Imatinib-resistant leukemia.

To examine the effects of TKIs on cancer stem cell (CSC) characteristics in K562 cells, we conducted a sphere formation assay. After 18 days of culture conducive to sphere formation (Figure 2H, Appendix A), K562-IR cells exhibited a higher sphere formation capacity than K562 cells, forming 126 ± 11.6 colonies compared to 87 ± 8.9 colonies (Figure 2I), respectively. Notably, under Imatinib treatment, K562-IR cells retained robust sphere-forming abilities compared to K562 cells. Specifically, in the presence of 0.5 and 1 μM of Imatinib, K562-IR cells formed 16.3 ± 5.6 and 35.3 ± 6.5 colonies, respectively, while K562 cells formed significantly fewer colonies, with counts of only 3.2 ± 1.3 and 5.1 ± 2.3 under the same conditions. Furthermore, K562-IR cells treated with Imatinib were observed to form numerous small cell spheres, a phenomenon not observed in Imatinib-treated K562 cells. Conversely, second-line TKIs, such as Dasatinib, Nilotinib, and Bosutinib, showed significant inhibition of CSC formation at drug concentrations of 0.5 and 1 μM. These findings not only underscore the enhanced CSC properties of K562-IR cells but also highlight the considerable inhibitory effects of second-line TKIs on CSC formation.

### 3.4. Gene Expression Profiling of Imatinib-Resistant K562 Cells

To identify potential BCR::ABL1 kinase domain mutations in our K562-IR cells, we utilized Sanger sequencing to compare the Ph’ junction of BCR12-ABL10 between K562 and K562-IR cells. The sequences showed 100% identity between the two cell lines. This result indicates that the drug resistance of K562-IR cells in this study is driven by mechanisms independent of BCR::ABL1 mutations. Subsequently, to uncover the signaling regulations contributing to Imatinib resistance, we performed RNA-seq gene expression analysis on both K562 parental and K562-IR cells (Figure 2J). The resulting volcano plot revealed significant gene expression differences, with 915 genes upregulated and 937 downregulated (criteria: log^2^ fold change > 1 and padj < 0.05) in K562-IR compared to K562 cells. Notably, among these differentially expressed genes (DEGs) (Figure 2K), the top 15 upregulated genes (red columns) in K562-IR are associated with cells stemness (*TMCC3*, *UGT2B7*, *PCAT18*, *SOX13*, *REG1A*), epithelial–mesenchymal transition (EMT) (*ACTA2*, *PECAM1*, *TEK*, *ITGA9*), inflammation (*PI16*), and metalloproteinase activity (*ADAM28*, *TIMP2*). Conversely, the downregulated DEGs (green columns) in K562-IR include genes related to cell fate determination (*SOX5*, *EPCAM*), cell movement (*ARHGAP4*, *PKP3*, *LAMA4*), cell cycle (*CCND2*), hemolytic disease (*CFH*, *CFHR1*), and neuronal functions (*GABRA2*, *SOX5*, *NLGN1*).

Further analysis using KEGG and GO databases categorized the significant DEGs identified. KEGG pathway analysis indicated that K562-IR cells predominantly exhibit enhanced cytokine–cytokine receptor interaction, hematopoietic cell lineage, and inflammatory responses (Appendix A). GO biological category analysis identified cytokine production and signaling as major functions activated in K562-IR cells (Appendix A), suggesting cytokine-mediated signaling as a crucial factor in the development of Imatinib resistance in CML. Additionally, the relevance of cancer-colony-forming ability in drug resistance was highlighted. Therefore, we selected two interleukin receptors (*IL6R*, *IL7R*) and one stemness-associated molecule (*MYC*), which were highly expressed in K562-IR, to investigate their roles in Imatinib resistance (Figure 2L). Western blot analysis confirmed that K562-IR cells exhibit significantly higher levels of IL6R, IL7R, and MYC proteins compared to K562 parental cells, aligning with our RNA-seq findings. These results underscore the potential of targeting cytokine signaling and stemness-associated molecules to overcome Imatinib resistance in CML.

### 3.5. CRISPR/Cas9-Mediated Targeting of Putative Genes Associated with Imatinib Resistance

To further investigate the functional roles of IL6R, IL7R, and MYC in K562-IR cells, we utilized the CRISPR/Cas9 system by designing two sgRNAs for each gene. All of the designed sgRNA protospacers target the positive strand, except *IL7R*_1, which targets the negative DNA strand allele (Figure 3C). Control experiments with K562-IR cells transduced with a scrambled (SC) target virus maintained the wild-type sequences for *IL6R*, *IL7R*, and *MYC*, as confirmed by Sanger sequencing, indicating no unintended gene editing. In contrast, the introduction of gene-editing constructs targeting *IL6R* (*IL6R*_1 and *IL6R*_2), *IL7R* (*IL7R*_1 and *IL7R*_2), and *MYC* (*MYC*_1 and *MYC*_2) into K562-IR cells resulted in significant gene disruptions at the expected cleavage sites (red arrow).

Tracking of Indels by Decomposition (TIDE) analysis revealed high gene editing efficiencies for both *IL6R* sgRNAs, with *IL6R_1* achieving a 97.6% efficiency (Figure 3A) and *IL6R_2* achieving an 84.9% efficiency (Figure 3B). The most common mutation for *IL6R* sgRNA_1 was a 1 bp deletion occurring in 58.4% of the edited cell pool, while *IL6R* sgRNA_2 resulted in a more diverse mutation spectrum, dominating 51.7% of the cells. For *IL7R* editing, sgRNA_2 (Figure 3D) demonstrated higher efficiency (97.4%) than *IL7R* sgRNA_1 (Figure 3C), with respective efficiencies of 60.4%. The *IL7R* sgRNA_1-edited cells primarily exhibited diverse mutations (33.7%), whereas *IL7R* sgRNA_2-edited cells mostly showed 2 bp deletions (32.8%). Editing efficiencies for *MYC* were robust as well, with *MYC* sgRNA_1 (Figure 3E) and *MYC* sgRNA_2 (Figure 3F) achieving editing rates of 83% and 89.8%, respectively. The predominant mutations were diverse in both *MYC*-edited cell pools, constituting 38.4% and 84.4% of the changes, respectively. Importantly, gene-editing predictions for *IL6R* (Appendix A), *IL7R* (Appendix A), and *MYC* (Appendix A) matched the observed patterns of genomic repair in K562-IR cells, primarily showcasing mutations at the designated cleavage sites, thus validating the targeted gene-editing approach.

### 3.6. Exploring the Role of IL6R, IL7R, and MYC in Conferring Imatinib Resistance to K562-IR Cells

To elucidate the significance of *IL6R*, *IL7R*, and *MYC* expressions in the context of Imatinib resistance in K562-IR cells, we assessed cell viability in the gene-edited cells described above. Results revealed that K562 cells were highly susceptible to Imatinib (Figure 4A), exhibiting an IC50 of 0.0798 μM, while the K562-IR cells had an IC50 of 0.711 μM. The *IL7R*-knockout (KO) K562-IR cells displayed a moderate increase in Imatinib sensitivity, with an IC50 of 0.5681 μM. However, *IL6R* KO and *MYC* KO K562-IR cells restored high sensitivity towards Imatinib, showing IC50 values of 0.0805 μM and 0.0865 μM, respectively. This suggests that the presence of *IL6R* and *MYC* in K562 is paramount for Imatinib resistance; targeting these proteins could potentially re-sensitize patients to Imatinib treatment. Subsequently, we investigated the potential of *IL6R*, *IL7R*, and *MYC* expressions in instigating cell apoptosis in K562-IR cells. Western blot analysis confirmed the absence of IL6R (Figure 4B), IL7R (Figure 4C), and MYC (Figure 4D) expressions in all gene-KO cells across both DMSO and Imatinib treatment groups. Critical apoptosis markers, namely PARP and caspase-3 cleavages, were remarkably upregulated in Imatinib-treated *IL6R* KO and *MYC* KO K562-IR cells compared to their DMSO-treated counterparts. Although *IL7R* KO led to an elevation in caspase-3 cleavage, it did not result in apoptosis in either treatment group. Flow cytometry reinforced these findings, indicating that Imatinib precipitated a substantial increase in the sub-G1 cell cycle phase, accounting for 8.82% of the K562-IR cell population (Figure 4E), in contrast to only 0.81% in the DMSO-treated group. Conversely, *IL6R* KO and *MYC* KO K562-IR cells under Imatinib treatment displayed significant sub-G1 cell populations, reaching 9.57% and 10.32% (Figure 4F), respectively, compared to a 1.55% baseline in the scrambled control. These data emphasize the critical roles of IL6R and MYC in maintaining the survival of K562-IR cells. Their elimination not only heightened the cells’ vulnerability to Imatinib-induced apoptosis but also enhanced the anticancer potency of the Imatinib treatment. Therefore, targeting IL6R and MYC presents a viable therapeutic strategy for treating patients with Imatinib-resistant CML, either alone or in conjunction with Imatinib therapy.

### 3.7. IL6R and MYC KO Reduces Colony-Forming Ability and Restores Imatinib Cytotoxicity

Next, we want to know the clinical potential of the NIADS v2 platform in evaluating apoptosis in *IL6R* and *MYC* KO K562-IR cells treated with DMSO or Imatinib. The bioluminescent signals, indicative of apoptosis, were significantly higher in *IL6R* and *MYC* KO cells compared to SC (scrambled control) and *IL7R* KO cells (Figure 5A, panels a and b, *p* ≤ 0.05), indicating that knockout of *IL6R* and *MYC* could induce cell apoptosis under both DMSO and Imatinib treatments. Furthermore, *IL6R* and *MYC* KO cells exhibited significantly enhanced cytotoxic response to Imatinib-induced apoptosis compared to the DMSO treatment group (*p* ≤ 0.05). This underscores the clinical potential of targeting *IL6R* and *MYC* to restore Imatinib sensitivity in patients resistant to the drug. Further assessments were conducted on colony-forming ability in *IL6R* and *MYC* KO K562-IR cells (Figure 5B). Tumor sphere formation assays indicated increased colony-forming ability, as observed in the number and size of spheres, in both K562-IR and SC cells (*p* ≤ 0.01) compared to K562 cells. Conversely, *IL6R* and *MYC* KO cells displayed a significant reduction in both number and size of spheres compared to SC cells (*IL6R* KO, *p* ≤ 0.001; *MYC* KO, *p* ≤ 0.01), suggesting reduced cancer malignancy. This diminished colony-forming ability was corroborated by real-time PCR analysis (Figure 5C), which revealed significantly lower gene expressions of CSC markers, such as *ALDH1*, *CD44*, and *CD47,* in *IL6R* and *MYC* KO cells compared to SC and parental K562-IR cells (*p* ≤ 0.01 or *p* ≤ 0.001) [28]. These findings suggest that *IL6R* and *MYC* act as key modulators contributing to cancer malignancy and enabling Imatinib resistance in CML, highlighting their potential as therapeutic targets.

## 4. Discussions

Beyond BCR::ABL1 mutations, genetic alterations have been critically evaluated for their role in conferring Imatinib resistance, especially in cancers like gastrointestinal stromal tumors (GISTs), where mutations in the KIT or PDGFRA genes are prevalent [29]. Clinical observations indicate that a substantial fraction of patients with PDGFRA-mutant and c-Kit-mutant GISTs exhibit resistance to Imatinib, with mutations, such as D842V in PDGFRA and D816V in c-Kit, being common in resistant cases [30]. Research has indicated that Dasatinib is more effective than Imatinib in countering these specific genetic alterations, underscoring its clinical value in treating PDGFRA mutant cases. Additionally, various mutations in c-Kit have been linked to different levels of resistance, underlining the necessity of genotyping KIT and PDGFRA to foresee therapeutic outcomes and inform the choice of second-line treatments for GIST patients [31]. Although KIT or PDGFRA mutations are infrequent in CML, there is emerging evidence that suggests a link between Imatinib resistance and mutations in the activation loops of these genes, signifying the wider importance of mutation profiling in tailoring treatment strategies across different cancer types [32].

Multispecific drug transporters play a pivotal role in the development of drug resistance in cancer cells, particularly as they acquire stemness characteristics, highlighting the complexity of addressing cancer at the molecular level [33]. The solute carrier (SLC) and ATP-binding cassette (ABC) transporter superfamilies are critical in the processes of drug absorption, distribution, metabolism, and elimination, marking their importance in the field of pharmacology [34]. Through next-generation sequencing (NGS) RNAseq analysis, we observed significant upregulation of stemness-associated genes (*TMCC3*, *UGT2B7*, *PCAT18*, *SOX13*, *REG1A*) among the top differentially expressed genes (DEGs) in K562-IR cells compared to their non-resistant counterparts [35,36,37]. Notably, *SLC25A48* and *SLC30A8* emerged as the most upregulated SLC transporters, with fold changes (log^2^) of 7.57 and 6.3, respectively, making them among the top 50 upregulated DEGs. This indicates that resistance mechanisms in K562-IR cells might shift away from traditional drug efflux transporters to variations in drug metabolism and elimination pathways. For instance, *UGT2B7* is implicated in drug deactivation and reduced drug sensitivity [38]. This transition to stemness-associated drug resistance mechanisms underscores the need for new therapeutic strategies that address these complex molecular changes in cancer cells.

Resistance to TKIs in CML presents a complex challenge extending beyond the previously emphasized point mutations in the BCR::ABL1 kinase domain that obstruct drug binding [39]. Notably, Cortes and colleagues have shown that approximately half of the clinical cases of TKI resistance do not involve mutations in the BCR::ABL1 kinase domain, pointing to kinase-independent mechanisms playing a significant role in the resistance observed in CML [40]. This assertion is bolstered by research highlighting the IL6/IL6R axis’s contribution to Imatinib resistance via activation of the JAK/STAT3 signaling pathway [41,42]. Moreover, the neutralization of IL6 in culture media markedly diminishes cell colony formation, emphasizing its essential function in maintaining malignant cell viability [43]. In parallel, the IL7/IL7R signaling pathway, traditionally linked to hematopoiesis, has been identified as a protector of K562 cells from Imatinib-induced apoptosis, mediated through the JAK1/STAT5 pathway and STAT5 phosphorylation [44]. Additionally, the oncogene *MYC*, associated with various cancer types, influences drug sensitivity in CML and aids in the survival of CML CD34+ leukemic stem cells [45]. MYC’s role extends to influencing the MAPK/HNRPK pathway [46] and regulating miR-150 [47], both implicated in myeloid differentiation and TKI resistance [48]. This extensive understanding emphasizes the multifaceted mechanisms of TKI resistance in CML, highlighting the necessity for targeted therapeutic approaches that address both BCR::ABL1 kinase-dependent and independent pathways to effectively overcome resistance.

Quantitative and kinetic evaluations of programmed cell death are essential in anti-cancer research, providing critical insights into how cells respond to stress and assisting in the development of effective drug screening techniques. Apoptosis, a fundamental form of cell death, is conventionally detected using certain methods, such as Western blot and Annexin V-binding assays, which primarily identify the early stages of apoptosis initiated by caspase activation [49]. The advent of high-content cell imaging technologies has significantly enhanced the study of cellular attributes, allowing for the real-time tracking of cellular dynamics via fluorescent markers [50]. These advancements are crucial for assessing cytotoxic responses to stressors, conducting genome-wide screenings, and refining drug screening processes. Nevertheless, these assays often depend on viability markers like propidium iodide (PI), DRAQ7, and SYTOX, which typically signal late-stage apoptotic events and fail to distinguish among various forms of cell death [51]. Moreover, the inconsistent use of these dyes can lead to over-labeling when there are disruptions in membrane integrity, highlighting the necessity for more accurate apoptosis detection methods capable of differentiating between the distinct stages and types of cell death.

In evaluating cell apoptosis, researchers frequently utilize probes labeled with reporters or designed for caspase cleavage, thus targeting proteins, such as caspase-3 (DEVD) or caspase-8 (IETD). However, issues, such as variable or diminished cleavage efficiency compared to natural caspase substrates, along with activation by non-caspase enzymes, have been identified, raising concerns about their reliability [52]. To overcome these challenges and address variations in cell growth and inconsistencies across wells that can affect assay results, some scientists have adopted secondary analysis methods like flow cytometry to normalize results between samples. Although this method can increase accuracy, it compromises the efficiency and simplicity sought in high-throughput live-cell imaging scenarios and is limited by the need for specific reporter constructs. To overcome these challenges, we introduced an advanced high-throughput bioluminescence-based NIADS platform for the simultaneous and precise quantification of cellular apoptosis and high-content drug screening in various cancer types [20,21,25,53]. This study further enhanced the NIADS v2 sensor by incorporating GFP expression. This improvement is designed to standardize data against fluctuations due to cellular behavior and assay conditions, thereby providing more reliable and unbiased apoptosis measurements.

## 5. Conclusions

In conclusion, this study aimed to refine the NIADS platform to expedite apoptosis detection, particularly for evaluating anti-cancer drugs within the scope of precision medicine (Figure 5D). By establishing an Imatinib-resistant K562 cell model, we successfully evaluated the cytotoxic impacts of second-line TKIs, such as Dasatinib, Nilotinib, and Bosutinib, utilizing the NIADS v2 alongside Western blotting for an in-depth analysis. Additionally, RNA-seq gene profiling combined with CRISPR/Cas9 gene-editing technology uncovered the crucial roles of IL6R and MYC in fostering cell survival, augmenting stemness features, and influencing Imatinib sensitivity in resistant variants. Our results indicate that K562 cells might engage alternative signaling pathways, such as PI3K/AKT, MAPK, and STAT5, thus circumventing BCR::ABL1 inhibition and maintaining CSC attributes and promoting Imatinib resistance. By spotlighting the importance of IL6R and MYC inhibition, our research proposes a new therapeutic approach to combat Imatinib-resistant CML, paving the way for novel interventions and potentially improving treatment efficacy for patients.

## Figures and Tables

**Figure 1 cells-13-00616-f001:**
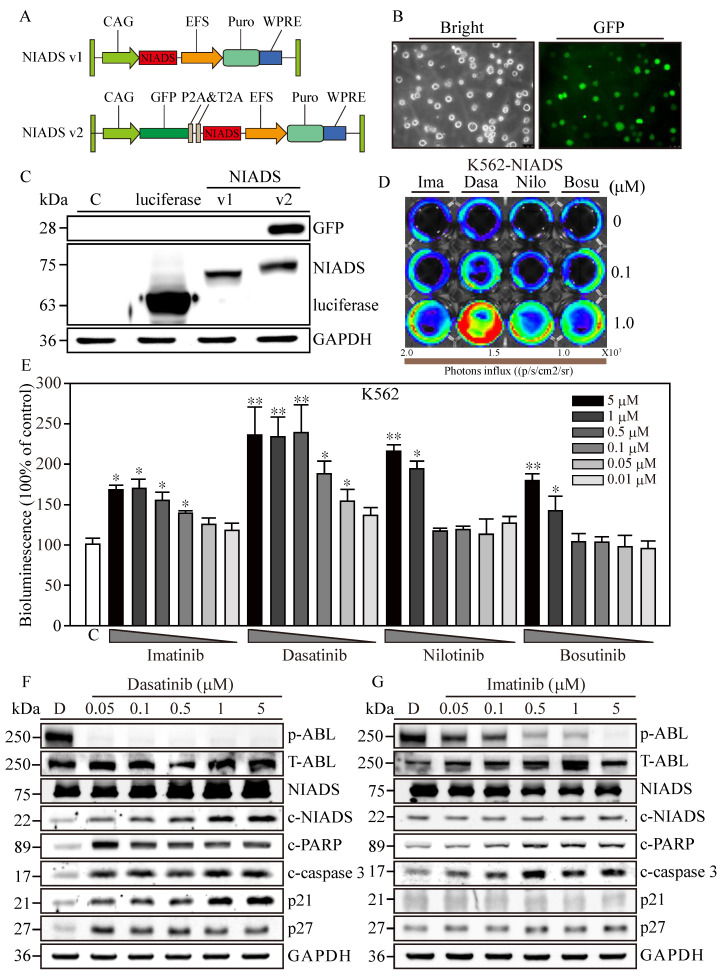
Optimization of the Non-Invasive Apoptosis Detection Sensor version 2 for cancer research applications. (**A**) The comprehensive schematic of the Non-Invasive Apoptosis Detection Sensor (NIADS) and its evolution into the GFP-integrated NIADS v2. The fusion of pepA-Nluc and pepBCluc fragments is intricately linked with three sequential DEVD caspase-3 cleavage sequences. As cellular apoptosis activates, the activated caspase-3 discerns the DEVD sequence, effectuating the cleavage of the fusion protein into two distinct fragments. Subsequently, the bioluminescence activity can be quantified with the introduction of the requisite substrate, Luciferin. To provide an accurate basis for normalization, the GFP signal is employed, which is essential for infection cell normalization. (**B**) Demonstration of NIADS v2 transfected cell images with both bright and GFP fluorescent illuminations. (**C**) The protein mass analysis encompasses luciferase, NIADS, and the NIADS v2 fusion proteins, each weighed meticulously. (**D**) IVIS image of NIADS v2 sensor; transfected K562 cells were subjected to varying concentrations (0, 0.1, and 1 μM) of first- and second-line BCR::ABL1 targeting agents, which encompass Imatinib, Dasatinib, Nilotinib, and Bosutinib. (**E**) The bioluminescence activity emanating from the NIADS v2 sensor demonstrates a discernible dosage-dependent response concerning the BCR::ABL1 targeting agents. All bioluminescent results were normalized with their internal GFP fluorescent activity. The potential luciferase activity inhibition was evaluated through Imatinib, Dasatinib, Nilotinib, and Bosutinib exposures (Appendix A). The *p* value ≤ 0.05 is presented as *, and ≤ 0.01 is presented as **. Western blot analysis involved varying concentrations of (**F**) Dasatinib- and (**G**) Imatinib-treated K562 cells. The expressions of NIADS fusion proteins are examined with full-length constructs as well as cleavage fragments. In addition, indicators of kinase inhibition were determined through ABL phosphorylation and apoptosis events were determined through cleaved PARP and cleaved caspase-3, whereas cell cycle regulation was examined according to p21 and p27 protein expressions. Western blot of NIADS v2 sensor expression and activation is shown in Appendix A. GAPDH serves as the loading control. Data are presented as the mean value with standard error.

**Figure 2 cells-13-00616-f002:**
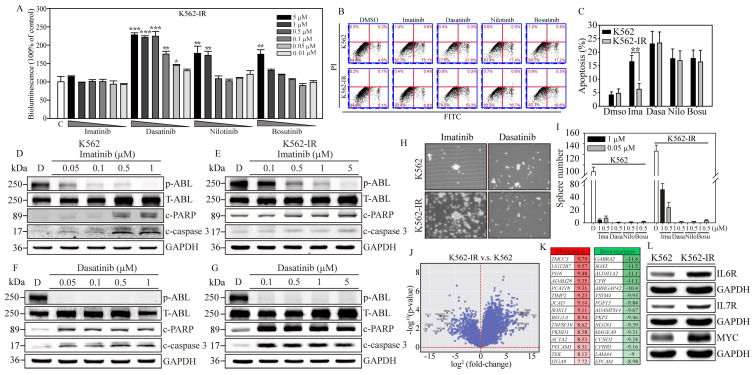
Gene expression profiling associated with Imatinib-resistant K562 cells. (**A**) The bioluminescence activity emanating from the NIADS v2 sensor demonstrates a discernible dosage-dependent response concerning the BCR::ABL1 targeting agents on K562-IR cells. All bioluminescent results were normalized with their internal GFP fluorescent activity. The *p* value ≤ 0.05 is presented as *, ≤0.01 is presented as **, and ≤0.001 is presented as ***. (**B**) The annexin V-FITC and PI flow cytometry were carried out to determine the K562 and K562-IR cell apoptosis population throughout 5 μM of various of TKI exposures for 24 h. (**C**) The apoptosis population was calculated and compared in each drug treatment group. The *p* value ≤ 0.01 is presented as **. Western blot analysis involved varying concentrations of Imatinib-treated (**D**) K562 cells and (**E**) K562-IR cells, and Dasatinib-treated (**F**) K562 cells and (**G**) K562-IR cells were determined for apoptosis induction. NGS RNA-Seq analysis on triplicate repeat of K562 and K562-IR cells. (**H**) The sphere formation assay was conducted to evaluate the CSC property throughout 1 μM of various of TKI exposures for 18 days on K562 cells and K562-IR cells. (**I**) The bar figure represents the colony number in each treatment group. (**J**) The differential gene expressions (DEGs) were presented in fold-change (log^2^) and significance (log^10^ *p*-value). (**K**) The top 15 most upregulated and downregulated DEGs are listed, and fold-changes are indicated. (**L**) The critical IL6R, IL7R, and MYC pathways were selected, and the key molecules were examined for protein induction. GAPDH serves as the loading control. Data are presented as the mean value with standard error.

**Figure 3 cells-13-00616-f003:**
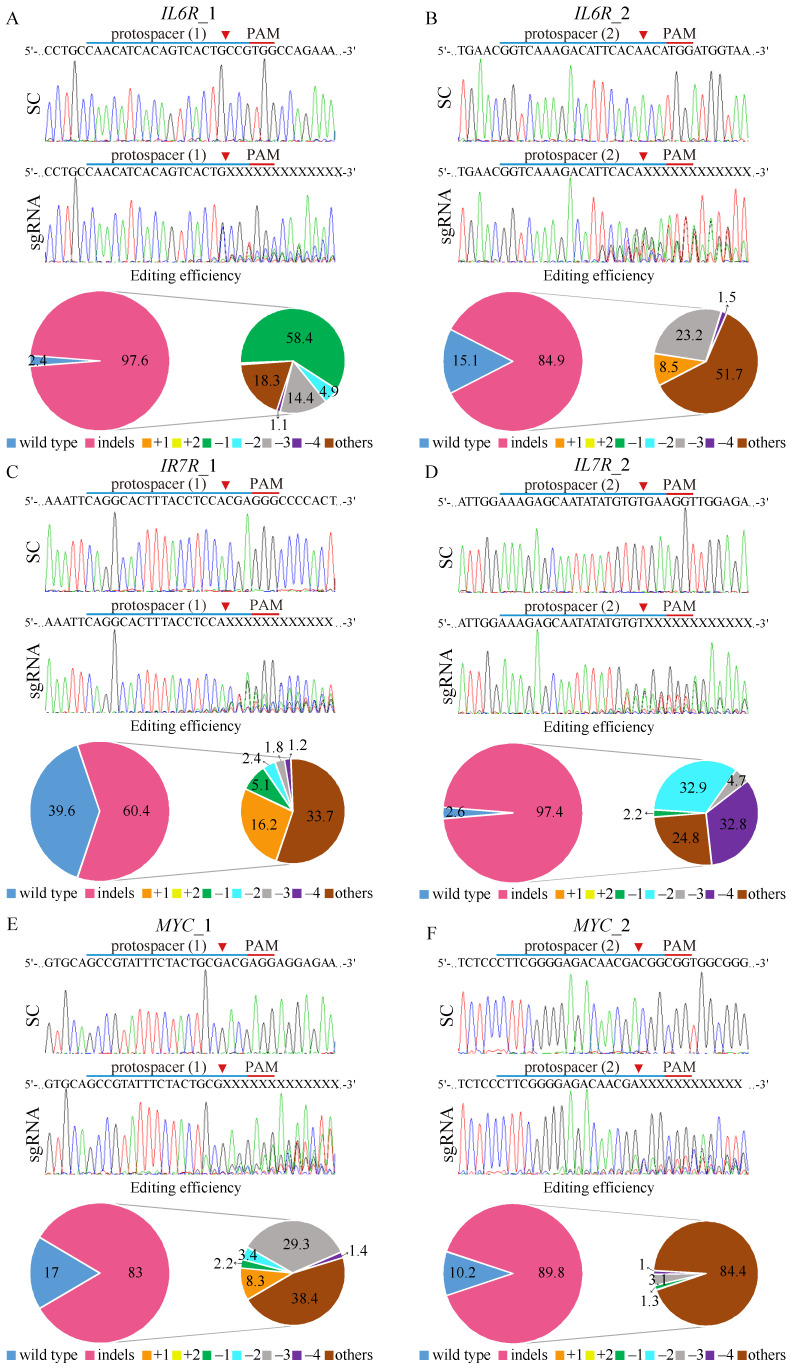
CRISPR/Cas9 gene editing of *IL6R*, *IL7R*, and *MYC* on K562-IR cells. The depiction of the human (**A**,**B**) *IL6R*, (**C**,**D**) *IL7R*, and (**E**,**F**) *MYC* DNA locus and corresponding protospacer sequences for gene editing, whereas the intended Cas9 cleavage site is indicated by an arrowhead. The protospacer adjacent motif (PAM), crucial for Cas9 nuclease activity, is highlighted in red. The strategy involves the delivery of scrambled (SC) sgRNA and gene targeting sgRNAs to K562-IR cells via lentivirus. Post-transduction, DNA from virus-infected cells is purified and subjected to Sanger sequencing, uncovering specific gene editing at the intended loci. Cells exposed to SC sgRNA exhibit wild-type sequences, while gene-targeting sgRNAs lead to a spectrum of sequences around the Cas9 cleavage point within a pool of gene-edited cells. The TIDE algorithm analysis quantifies gene-edited sequences (indels, insertions, and deletions), revealing a high editing efficiency in K562-IR cells. The left pie charts display overall gene editing efficiency in the cell pool, while the specific indels are detailed in the right pie charts.

**Figure 4 cells-13-00616-f004:**
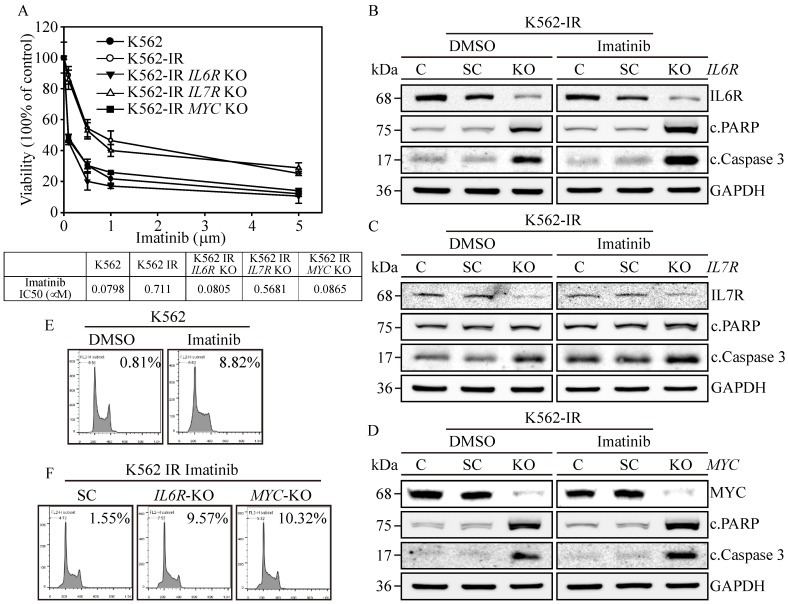
Biological impacts on *IL6R*, *IL7R*, and *MYC* gene-edited K562-IR Cells. (**A**) The cell viability was determined through MTT assay. *IL6R*, *IL7R*, and *MYC* KO K562-IR cells were subjected to varying concentrations of Imatinib, with their respective IC50 values measured. The data are depicted as the mean value alongside the standard error. Apoptosis induction in (**B**) *IL6R*, (**C**) *IL7R*, and (**D**) *MYC* KO K562-IR cells was protein-analyzed through Western blot assay with DMSO or 1 μM of Imatinib treatments. GAPDH served as the loading control. Flow-cytometry-based examination of the sub-G1 cell cycle phase was used on (**E**) DMSO and Imatinib treated on K562 cells. Lastly, (**F**) the SC, *IL6R*, and *MYC* KO K562-IR cells with 1 μM of Imatinib treatments were determined for the sub-G1-phase population.

**Figure 5 cells-13-00616-f005:**
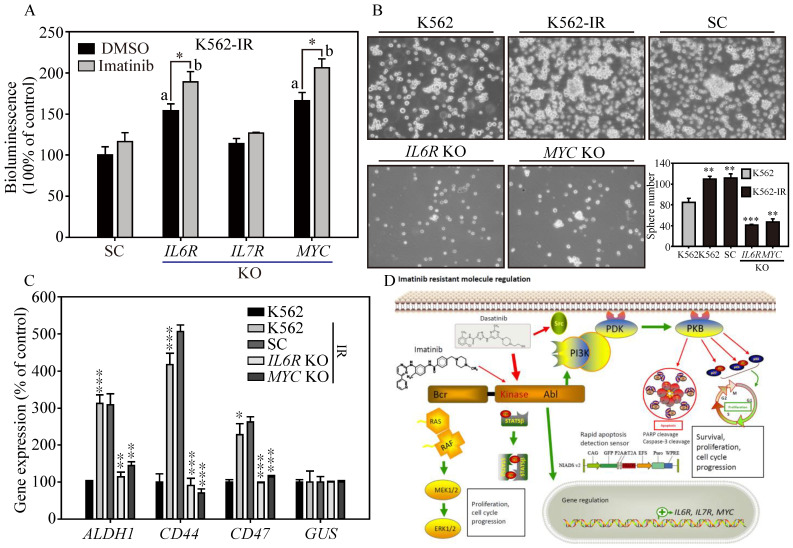
The cancer malignancy of *IL6R* and *MYC* KO in K562-IR cells. (**A**) NIADS-v2-based assay was employed to assess apoptosis in *IL6R*, *IL7R*, and *MYC* KO K562-IR cells exposed to DMSO or 1 μM of Imatinib. All bioluminescent readings were normalized to the internal cellular GFP fluorescence. Statistical significance with a value less than 0.05 is indicated with “a” for gene KO versus SC comparisons and “b” for DMSO versus Imatinib treatment comparisons. (**B**) Sphere formation assays were conducted to examine the cancer malignancy of *IL6R* and *MYC* KO cells derived from K562-IR cells. Both images and quantitative data of the resulting cell spheres are provided. (**C**) The gene expressions of CSC markers, such as *ALDH1*, *CD44*, and *CD47*, examined in the above cells are illustrated. The data are presented as mean ± standard errors. Statistical differences between groups were evaluated using Student’s *t*-test, with *p*-values denoting significance levels; *p*-value ≤ 0.05 is indicated by *, *p*-value ≤ 0.01 is represented by **, and *p*-value ≤ 0.001 is marked by ***. (**D**) A schematic diagram summarizing the key findings of this research is provided, illustrating the study’s major conclusions and contributions to the understanding of stemness and drug resistance in cancer cells.

## Data Availability

The data presented in this study are available upon request from the corresponding author.

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
