# Peer review of "Unveiling IL6R and MYC as Targeting Biomarkers in Imatinib-Resistant Chronic Myeloid Leukemia through Advanced Non-Invasive Apoptosis Detection Sensor Version 2 Detection"

_cells, 2024, doi:10.3390/cells13070616_

Round 1
Reviewer 1 Report (Previous Reviewer 2)
Comments and Suggestions for Authors
Summary: In the present study, the authors assessed the role of IL6R, IL7R, and MYC genes in imatinib resistance of the K562 CML cell line. Altogether, their data reinforce the findings of other groups suggesting that IL6 and MYC play a role in the mechanism(s) underlying BCR::ABL1 kinase-independent imatinib resistance in K562 cells. The use of a novel Non-Invasive Apoptosis Detection Sensor version 2 (NIADS v2) improves the novelty of the study. The revised manuscript is nicely written, with only minor comments for improvement.
MINOR:
1) Lines 72-73: Remove “This necessitates the exploration of alternative therapeutic approaches.” This statement is a repeat from the previous sentence.
2) The numbers in the dot plots of Figure 2B are illegible.
3) Line 377: FTIC should be FITC.
4) It is not appropriate to talk about ‘stemness’ when only using K562 cells. These are cell lines and therefore cannot be used to study stem cells. Please consider changing the word ‘stemness’ to ‘survival advantage’ or ‘colony forming ability.’
Author Response
Reviewer 1:
In the present study, the authors assessed the role of IL6R, IL7R, and MYC genes in imatinib resistance of the K562 CML cell line. Altogether, their data reinforce the findings of other groups suggesting that IL6 and MYC play a role in the mechanism(s) underlying BCR::ABL1 kinase-independent imatinib resistance in K562 cells. The use of a novel Non-Invasive Apoptosis Detection Sensor version 2 (NIADS v2) improves the novelty of the study. The revised manuscript is nicely written, with only minor comments for improvement.
1. Lines 72-73: Remove “This necessitates the exploration of alternative therapeutic approaches.” This statement is a repeat from the previous sentence.
Response: Thank you for pointing out the redundancy. We have removed the repeated sentence as suggested. Please refer to lines 72-73 in the revised manuscript for this change.
2. The numbers in the dot plots of Figure 2B are illegible.
Response: Thank you for your feedback. We have ensured that the percentage of the cell population in Figure 2B is clearly labeled. Please refer to the revised Figure 2B for these updates.
3. Line 377: FTIC should be FITC.
Response: Thank you for pointing out the typo. It has been corrected.
4. It is not appropriate to talk about ‘stemness’ when only using K562 cells. These are cell lines and therefore cannot be used to study stem cells. Please consider changing the word ‘stemness’ to ‘survival advantage’ or ‘colony forming ability.’
Response: Thank you for highlighting this issue. I concur with your comment and have accordingly replaced the term ‘stemness’ with more suitable expressions like ‘colony forming ability’ and ‘cancer malignancy’ throughout the manuscript. These modifications are highlighted in the revised document for easy identification.
Reviewer 2 Report (New Reviewer)
Comments and Suggestions for Authors
Lee and coworkers used imatinib-resistant K562 cells to identify novel mechanisms of treatment-resistance. The approach is interesting and reveales potential targets which may aid to establish novel therapeutic treatments.
I have found several points which require correction:
1. The text lacks several citations, including the first sentence of the introduction (line 51), CML in line 65, robust apoptosis in line 114, CSC markers in line 543, stemness genes in line 599.
2. It is important to use authenticated cell lines. You have used K562 cells which were (unauthorized) obtained from a colleague. Please perform authentication of your cell line and add the results in the supplement.
3. The materials and method section lacks several informations: company name for cell line Phoenix-ECO, company names for all chemicals, medium supplements and drugs including TKIs, for bioluminescence imaging device, for ImageJ.
4. Please explain the function of your established method using NIADS in the introduction or in the results section.
5. Delete lines 275 to 286 - it is a repetition.
6. Please discuss or comment the advantage of your new method for suspension cells. You mention in line 343 that K652 cells are hard to transfect. Thus, you have to perform lentiviral transduction to perform your detection system instead of traditional apoptosis assays which work reliable, directly and fast.
7. I am not convinced by the sphere formation assay which lasts 18 days. Do you replenish medium and drugs during that long time?
8. The used methods of Sanger sequencing and RNA-seq are not described in the methods section. The methods section also lacks the description and according citations of performed volcano plot, KEGG and GO databases, the performance of the CRISPR/CAS9 assay, and the TIDE analysis.
9. The results shown in Figure 3 are technical data and may be shifted in the supplement.
10. The reference list does not correspond to the style used for this journal.
Author Response
Reviewer 2:
Lee and coworkers used imatinib-resistant K562 cells to identify novel mechanisms of treatment-resistance. The approach is interesting and reveales potential targets which may aid to establish novel therapeutic treatments.
I have found several points which require correction:
1. The text lacks several citations, including the first sentence of the introduction (line 51), CML in line 65, robust apoptosis in line 114, CSC markers in line 543, stemness genes in line 599.
Response: Thank you for your observations regarding the need for additional citations. We have carefully reviewed the manuscript and now highlighted and appropriately cited all references, including those mentioned in your feedback. Please refer to the updated sections for the incorporated references.
2. It is important to use authenticated cell lines. You have used K562 cells which were (unauthorized) obtained from a colleague. Please perform authentication of your cell line and add the results in the supplement.
Response: Thank you for emphasizing the importance of using authenticated cell lines. Given the tight deadline for this revision round (5 days), I have expedited the process by sending the K562 cells for authentication at a biotechnology service, which is expected to take approximately 3 days. I assure you that the certificate of authentication will be included in the supplementary information of this study upon receipt.
3. The materials and method section lacks several informations: company name for cell line Phoenix-ECO, company names for all chemicals, medium supplements and drugs including TKIs, for bioluminescence imaging device, for ImageJ.
Response: Thank you for pointing out the missing details. We have updated the materials and methods section to include catalog numbers, company names, and locations for all specified items, such as the Phoenix-ECO cell line, chemicals, medium supplements, drugs including TKIs, the bioluminescence imaging device, and ImageJ software. Please refer to the revised section for this comprehensive information.
4. Please explain the function of your established method using NIADS in the introduction or in the results section.
Response: Thank you for your suggestion. We have included a concise explanation of the NIADS mechanism at the beginning of the results section. Please refer to lines 272-248 in the revised manuscript for this addition.
5. Delete lines 275 to 286 - it is a repetition.
Response: Thank you for pointing out the repetition. It has been removed from the revised manuscript.
6. Please discuss or comment the advantage of your new method for suspension cells. You mention in line 343 that K652 cells are hard to transfect. Thus, you have to perform lentiviral transduction to perform your detection system instead of traditional apoptosis assays which work reliable, directly and fast.
Response: Thank you for your comment. I apologize for any lack of clarity regarding this point. It is well recognized that transfecting suspension cells, like K562 cells, poses challenges. To ensure efficient gene delivery, we utilized lentivirus transfection, enhanced by additional spin infection, to increase transfection efficiency in K562 cells. We then compared the sensitivity of traditional apoptosis assays to our NIADS v2 system. The results demonstrated that NIADS v2 provides more reliable, accessible, and unbiased apoptosis measurements compared to traditional methods. Furthermore, the advantages of the NIADS v2 sensor are discussed in detail towards the end of the discussion section. Please refer to lines 667 to 673 for this elaboration.
7. I am not convinced by the sphere formation assay which lasts 18 days. Do you replenish medium and drugs during that long time?
Response: Sphere formation assays are widely used to assess stemness characteristics and cancer malignancy in cancer cells. In our experiment, a minimal number of cells (10,000 cells/well) were cultured in a 6-well ultra-low attachment plate, which, coupled with serum-free medium supplemented appropriately, suffices for cancer cells to form spheres without the need for medium and drug replenishment. Our sphere formation methodology is based on protocols from established research, such as the one described in Cancer Cell International, volume 15, Article number: 95 (2015), which outlines a comprehensive procedure for conducting such assays, including the use of DMEM/F12 medium supplemented with 1% B27 supplement, 0.01% antibiotic–antimycotic, 20 ng/mL epidermal growth factor [EGF], and 20 ng/mL basic fibroblast growth factor [bFGF], all conducive to sphere formation in ultra-low attachment conditions. This reference supports our methodology and demonstrates the robustness of our approach in investigating the stemness property and malignancy of cancer cells under specified culture conditions. We added a reference to support this sphere formation protocol. Please see the reference in line 252.
8. The used methods of Sanger sequencing and RNA-seq are not described in the methods section. The methods section also lacks the description and according citations of performed volcano plot, KEGG and GO databases, the performance of the CRISPR/CAS9 assay, and the TIDE analysis.
Response: Thank you for your attention to detail. We have expanded the materials and methods section to include detailed descriptions of "Sanger Sequencing and Gene Editing Efficiency Assay" and "RNA Library Preparation, Sequencing, and Analysis." These additions aim to comprehensively detail the RNA sequencing and CRISPR gene editing analysis methodologies used in our study. Please refer to the highlighted sections on lines 158-168 and 184-203 in the revised manuscript for these updates.
9. The results shown in Figure 3 are technical data and may be shifted in the supplement.
Response: Thank you for your suggestion. I appreciate the reviewer's perspective on this matter. Nonetheless, the CRISPR gene editing of MYC, IL6R, and IL7R represents a pivotal and crucial component of our study, as it directly demonstrates the significant roles of MYC, IL6R, and IL7R in sustaining drug resistance and cancer malignancy within K562-IR cells. Furthermore, the technical data related to Figure 3 have already been included in Supplementary Figures 5-7. Given the centrality of these gene editing results in validating our study's hypotheses, I believe it is essential to showcase the DNA disruptions and gene indels at the MYC, IL6R, and IL7R loci following CRISPR editing within the main body of the manuscript.
10. The reference list does not correspond to the style used for this journal.
Response: Thank you for highlighting this discrepancy. I have followed the reference style as outlined in the “Instructions for Authors” on the Cells journal website (Cells | Instructions for Authors - mdpi.com). If there are any remaining discrepancies in the reference list style, I kindly request assistance from the editors for the necessary revisions during manuscript proofreading process.
This manuscript is a resubmission of an earlier submission. The following is a list of the peer review reports and author responses from that submission.
Round 1
Reviewer 1 Report
Comments and Suggestions for Authors
In the present study, the authors have optimized apoptosis detection sensor, Non-Invasive Apoptosis Detection Sensor version 2 (NIADS v2) by using CRISPR/Cas9 gene editing and have elucidated that increased expression of IL6R, IL7R, and MYC genes are the underlying mechanisms of imatinib resistance development in K562-IR cell line. I have following concerns.
1) In figure 1D and 1E) the authors have shown the dose dependent effect of different 1st and 2nd generation tyrosine kinase inhibitors (TKIs). Authors shall include statistical analysis if there is any significant difference in the Bioluminescence observed among groups.
2) The dose dependent effect of Dasatinib on Bioluminescence is clearly very robust as shown in Figure 1E, but it is not matching with its respective Western Blot data (indicators of apoptosis). However, the dose dependent effect of Imatinib on Bioluminescence was not very profound but in WB the effect is very evident. Could you please explain the possible causes?
3) In Figure 2A. Dasatinib is inducing apoptosis evidently as shown in the Figure but there is no effect of Imatinib. It would be great if authors could explain whether the effects are statistically significant or not. Also the western blot image do not match with bioluminescence data (Fig 2C)
4) It would be interesting if authors could have looked at the colony forming abilities of both K562 and K562-IR cells in presence of different tyrosine kinase inhibitors.
5) Please include the densitometric analysis for all the western blots.
6) Based on the RNA-seq data was there any difference observed in the expression of Drug efflux transporters genes.
7) What would have been the consequence of knocking down IL6R, IL7R, and MYC simultaneously in K562-IR cells and treating the cells with varying concentrations of TKIs.
8) Have the authors also looked at the expression of IL6R, IL7R, and MYC in CML patient samples.
9) What are the upstream regulators of the differentially expressed genes in K562-IR vs K562?
10) Have the authors also checked apoptosis via Annexin V/PI in K562-IR vs K562 in presence of TKIs?
Comments on the Quality of English LanguageSlight work on grammar is needed
Author Response
Reviewer#1
In the present study, the authors have optimized apoptosis detection sensor, Non-Invasive Apoptosis Detection Sensor version 2 (NIADS v2) by using CRISPR/Cas9 gene editing and have elucidated that increased expression of IL6R, IL7R, and MYC genes are the underlying mechanisms of imatinib resistance development in K562-IR cell line. I have following concerns.
1) In figure 1D and 1E) the authors have shown the dose dependent effect of different 1st and 2nd generation tyrosine kinase inhibitors (TKIs). Authors shall include statistical analysis if there is any significant difference in the Bioluminescence observed among groups.
Response: Thank you for your valuable feedback regarding Figure 1D and 1E in our manuscript. We acknowledge the importance of demonstrating statistical significance in our findings related to the dose-dependent effects of various 1st and 2nd generation tyrosine kinase inhibitors (TKIs). To address this, we have included a comprehensive statistical analysis in the revised version of our manuscript.
Specifically, the statistical significances pertaining to the differences in bioluminescence observed among different groups have been thoroughly analyzed and are now clearly presented in Figure 1E. For your convenience, we have also highlighted the relevant findings and their statistical implications in the figure legend. Please refer to the revised Figure 1E and the accompanying figure legend on page 7, line 15 for a detailed understanding of these enhancements.
We believe that these modifications and additions provide a clearer and more robust representation of our research findings, further substantiating our conclusions. We appreciate your attention to this detail and hope that the revised content adequately addresses your concerns."
2) The dose dependent effect of Dasatinib on Bioluminescence is clearly very robust as shown in Figure 1E, but it is not matching with its respective Western Blot data (indicators of apoptosis). However, the dose dependent effect of Imatinib on Bioluminescence was not very profound but in WB the effect is very evident. Could you please explain the possible causes?
Response: Thank you for your insightful comment regarding the correlation between the bioluminescence data and Western Blot results in our study, particularly concerning Dasatinib and Imatinib treatments on K562 cells.
In Figure 1E, we observed a robust induction of apoptosis, as indicated by bioluminescence, in response to Dasatinib treatment, with statistically significant effects noted from as low as 0.5 µM at the 8-hour mark. Correspondingly, Figure 1F presents significant caspase 3 activation following 0.5 µM Dasatinib treatment at 24 hours, which aligns with the expected progression of apoptotic events. Conversely, while Imatinib's effect on apoptosis bioluminescence was not as pronounced at the 8-hour mark with 1 µM treatment, Figure 1G clearly demonstrates significant caspase 3 activation at the 24-hour time point, indicating a delayed but evident apoptotic response. These observations suggest that while both Dasatinib and Imatinib effectively induce apoptosis in K562 cells, the temporal dynamics of their action differ. This is further evidenced by the sensitivity of our NIADS v2 system, which detected early apoptotic events at the 8-hour mark, in contrast to the Western Blot analysis which was more reflective of the complete apoptotic process at 24 hours.
Therefore, the apparent discrepancy between the bioluminescence and Western Blot data can be attributed to the different sensitivities of these assays in detecting early versus late-stage apoptosis, influenced by the duration of drug exposure. Our findings highlight the nuanced nature of apoptotic responses to Dasatinib and Imatinib and underscore the importance of considering time-dependent effects in drug response studies.
3) In Figure 2A. Dasatinib is inducing apoptosis evidently as shown in the Figure but there is no effect of Imatinib. It would be great if authors could explain whether the effects are statistically significant or not. Also the western blot image do not match with bioluminescence data (Fig 2C).
Response: Thanks again for this comment. The significances have been addressed in figure 2A. Please check the figure 2A and highlighted figure legend in page 10, line 5-6 in the revised manuscript. Regarding to the second question of Fig 2E, caspase 3 cleavage is weakly expressed in Imatinib treated K562-IR cells. We repeated a western blotting of c-caspase-3 in Figure 2E to demonstrate the drug resistance in K562-IR. Please check the Figure 2E in the revised manuscript and Supplementary information.
4) It would be interesting if authors could have looked at the colony forming abilities of both K562 and K562-IR cells in presence of different tyrosine kinase inhibitors.
Response: We thank the reviewer for their meaningful suggestion regarding the assessment of colony forming abilities of K562 and K562-IR cells in the presence of different tyrosine kinase inhibitors (TKIs).
Experimental Findings:
In response to this suggestion, we have conducted experiments that are detailed in our manuscript. Our findings demonstrate that K562-IR cells exhibit a significantly higher number of sphere colonies compared to K562 cells, a trend that persists even with Imatinib treatment. Furthermore, when exposed to different TKIs at concentrations of 0.5 or 1 μM, both cell types showed substantial inhibition of cancer stem cell (CSC) formation.
Significance of Results:
These results are crucial as they not only highlight the enhanced CSC properties in K562-IR cells but also reveal the significant CSC-inhibitory effects of second-line TKIs. This is particularly relevant in the context of drug resistance and the search for effective therapeutic strategies.
Location of Results in Manuscript:
We invite the reviewer to refer to Figure 2H-2I and Supplementary Figure 3 for representation of these findings. A detailed discussion of these results can be found in the results section (page 8, lines 37-50) in the revised manuscript.
We believe that these additional data provide a comprehensive understanding of the colony-forming abilities of these cells under different TKI treatments and add significant value to our study.
5) Please include the densitometric analysis for all the western blots.
Response: We thank the reviewer for their valuable suggestion to include densitometric analysis for all western blots presented in our study. Ensuring comprehensive and quantifiable data is essential for the validation of our results.
Inclusion of Densitometric Analysis:
In response to the reviewer’s recommendation, we have conducted a comparative analysis of band intensity across all western blots. This analysis provides a quantifiable measure of protein expression levels, enhancing the robustness of our data interpretation.
Placement in Supplementary Information:
To maintain the readability of the main text and not overwhelm the readers with excessive detail, we have chosen to place the protein expression percentage data in the Supplementary Information. This approach ensures that the main manuscript remains focused and accessible, while still providing detailed analysis for those interested.
Ensuring Accessibility and Clarity:
We believe this approach maintains the readability of the main text while ensuring that all relevant quantitative data is readily accessible for those interested in a more detailed analysis.
We hope that this addition addresses the reviewer’s concerns effectively and contributes to a more thorough presentation of our research findings.
6) Based on the RNA-seq data was there any difference observed in the expression of Drug efflux transporters genes.
Response: Thank you for highlighting the importance of exploring the role of drug efflux transporter genes, especially in the context of Imatinib resistance. We appreciate this insightful suggestion and have indeed addressed this aspect in our study.
In the revised manuscript, we have expanded the discussion section to include a detailed analysis of drug efflux transporter genes based on our RNA-seq data. This includes an examination of their expression patterns and potential implications in mediating Imatinib resistance. Please refer to the enhanced discussion on page 16, lines 3-15, where we have thoroughly discussed these findings. To support our analysis and provide a robust context, we have also incorporated additional references (numbered 32-34 in the revised manuscript). These references have been specifically chosen to enrich our discussion and provide a comprehensive understanding of the role of drug efflux transporters in the context of our study.
We believe that this additional analysis significantly contributes to the overall depth and scientific rigor of our research, offering valuable insights into the mechanisms of drug resistance in cancer therapy. We hope that the revised section effectively addresses your query and adds to the understanding of this complex interaction
7) What would have been the consequence of knocking down IL6R, IL7R, and MYC simultaneously in K562-IR cells and treating the cells with varying concentrations of TKIs.
Response: Thank you for posing this insightful question regarding the simultaneous knockdown of IL6R, IL7R, and MYC in K562-IR cells and the subsequent treatment with various concentrations of tyrosine kinase inhibitors (TKIs).
In this manuscript, our primary focus was on identifying key genes that contribute to drug resistance in K562 cells. As illustrated in Figure 5A, we demonstrated that the knockout (KO) of IL6R and MYC significantly attenuated the resistance to Imatinib in K562-IR cells. This finding underscores the potential of targeting IL6R and MYC as strategies to either prevent the emergence of Imatinib resistance or to resensitize K562-IR cells to Imatinib treatment. Regarding the impact of simultaneous IL6R, IL7R, and MYC knockdown on the efficacy of second-generation TKIs, our current study did not directly investigate this scenario. However, based on the data presented, we hypothesize that such a triple knockdown could potentially enhance the sensitivity of K562-IR cells to second-generation TKIs. This hypothesis is supported by the findings presented in Figure 2A, where the second-generation TKIs demonstrated a modulatory effect and potentially better therapeutic outcomes. Future studies could be designed to specifically investigate the combined effect of IL6R, IL7R, and MYC knockdown on the response to various TKIs. Such research would provide valuable insights into the intricate molecular mechanisms governing TKI resistance and help in the development of more effective therapeutic strategies for treating Imatinib-resistant cases.
We appreciate your question as it highlights an important area for future investigation, and we look forward to exploring these aspects in subsequent studies.
8) Have the authors also looked at the expression of IL6R, IL7R, and MYC in CML patient samples.
Response: Thank you for your query regarding the examination of IL6R, IL7R, and MYC expression in chronic myeloid leukemia (CML) patient samples. Your question highlights a critical aspect of translational research in leukemia.
While we acknowledge the importance of validating our findings in clinical samples, CML, being a relatively less common subtype of leukemia, poses certain challenges in terms of data availability. In this instance, we encountered limitations in accessing comprehensive CML patient data from large public databases like The Cancer Genome Atlas (TCGA) or cBioPortal, which typically serve as robust resources for such analyses. Given these constraints, our study did not include an analysis of IL6R, IL7R, and MYC expression in CML patient samples. However, we recognize the value such data would bring to substantiating our findings and understanding the clinical relevance of these markers in CML. We suggest that future studies with access to clinical CML samples could further explore these aspects. Investigating these markers in the context of CML patient samples would provide invaluable insights into their role and therapeutic potential in CML treatment.
We appreciate your suggestion as it underscores the necessity of bridging the gap between laboratory findings and clinical applications in leukemia research.
9) What are the upstream regulators of the differentially expressed genes in K562-IR vs K562?
Response: Thank you for your inquiry about the upstream regulators of the differentially expressed genes (DEGs) observed in the comparison between K562-IR and K562 cells.
As outlined in our manuscript, Imatinib resistance in chronic myeloid leukemia can arise through a variety of mechanisms. These include both BCR::ABL1-dependent and BCR::ABL1-independent pathways, as detailed in the discussion section (page 16, lines 16-31). The complexity of these resistance mechanisms encompasses a wide array of factors, such as transcription factors, signal transduction molecules, regulators of the cell cycle, metabolic pathways, autophagy processes, and elements related to the niche microenvironment. Given the multifactorial nature of Imatinib resistance and the extensive network of potential regulatory elements involved, pinpointing specific upstream regulators solely based on DEGs from RNA sequencing data presents a significant challenge. The current RNAseq analysis in our study primarily focused on identifying gene expression changes in K562-IR cells compared to K562 cells (figure 2J-K), which provides valuable insights into the molecular alterations associated with resistance. However, delineating the exact upstream regulatory mechanisms from this data alone is beyond the scope of our current investigation. We acknowledge the importance of understanding the upstream regulatory dynamics in Imatinib resistance and suggest that this constitutes an important area for future research. Advanced studies employing integrative approaches combining RNAseq data with other molecular and cellular analyses would be required to comprehensively identify and validate the upstream regulators contributing to Imatinib resistance.
We appreciate your question as it highlights a key area for further exploration in the field of leukemia research
10) Have the authors also checked apoptosis via Annexin V/PI in K562-IR vs K562 in presence of TKIs?
Response: We appreciate the reviewer's inquiry about the evaluation of apoptosis in K562-IR compared to K562 cells in the presence of tyrosine kinase inhibitors (TKIs).
Apoptosis Evaluation Methodology:
In response to this query, we have indeed analyzed apoptosis events using Annexin V/Propidium Iodide (PI) staining in both K562-IR and K562 cell lines when treated with TKIs. This assessment was conducted through flow cytometry analysis, allowing for a precise and quantitative evaluation of apoptotic cell populations.
Results and Findings:
The results of this analysis are presented in Figure 2B-2C of our revised manuscript. These figures provide a clear visual representation of the differential apoptotic responses between K562-IR and K562 cells under TKI treatment conditions.
Location of Detailed Discussion in Manuscript:
For a more comprehensive understanding of these findings, we invite the reviewer to refer to the Results section, specifically page 8, lines 12-21 of our revised manuscript. This section offers an in-depth discussion of the apoptosis results, their implications, and how they contribute to our understanding of the response of these cell lines to TKI treatment.
We believe that these results significantly contribute to the manuscript by providing a deeper insight into the differential responses of K562-IR and K562 cells to TKIs, particularly in the context of apoptosis induction.
Reviewer 2 Report
Comments and Suggestions for Authors
Summary: In the present study, the authors assessed the role of IL6R, IL7R, and MYC genes in imatinib resistance of the K562 CML cell line. Altogether, their data reinforce the findings of other groups suggesting that IL6 and MYC play a role in the mechanism(s) underlying BCR::ABL1 kinase-independent imatinib resistance in K562 cells. The use of a novel Non-Invasive Apoptosis Detection Sensor version 2 (NIADS v2) improves the novelty of the study. Specific comments to improve the manuscript are included below:
MAJOR:
1) Introduction, 1st Paragraph: When talking about “the presence of specific mutations” it would be helpful to explain that these mutations block drug binding of these drugs to the ATP binding site, thereby blocking their activity.
2) The Introduction could be improved by introducing the following changes:
a. The 1st and 2nd paragraphs are devoted entirely to CML, then the 3rd paragraph speaks more broadly about apoptosis and cancer. It feels disjointed. Perhaps the 3rd paragraph should come 1st, followed by a condensed 2nd paragraph talking about CML. This way it starts more broad and becomes more specific.
b. IL6R, IL7R, and MYC come out of nowhere in the final paragraph. Perhaps the authors could provide a little more information regarding the known role of these pathways in TKI resistance, and why you chose to study them here:
i. PMID: 16293596
ii. PMID: 27281222
iii. PMID: 25965572
iv. PMID: 30049824
v. PMID: 33712704
vi. PMID: 36536477
3) How many replicates were performed for each experiments? This should be included in the Statistical Analysis section and/or figure legends.
4) Figure 1, Panel E: Please include statistics on the graph or in the figure legend.
5) Do the K562-IR cells demonstrate BCR::ABL1-dependent or –independent resistance? Do they harbor BCR::ABL1 kinase domain mutations?
6) The immunoblot data throughout the manuscript should include BCR::ABL1 tyrosine kinase activity, either through using anti-pABL1, anti-phosphotyrosine, or anti-pCRKL antibodies compared total controls. This data will easily answer the previous question.
7) Figure 2, Panel A: Please include statistics on the graph or in the figure legend.
8) The paragraph under “Synopsis of Research Discoveries” should really be the first paragraph of the discussion, rather than a subsection under Results.
9) The structure of the abstract, introduction, and discussion sections should all be relatively similar. Please revise accordingly:
a. Novel apoptosis detection system
b. Applying this system to CML
c. Looking at IL6R, IL7R, and MYC in TKI resistance using our novel system.
MINOR:
10) Line numbers would make this manuscript easier to review.
11) BCR-ABL1 should be replaced with BCR::ABL1 throughout the manuscript.
12) Results Title: “Imatinib-Resistant associate gene expression profiling” should be changed to “Gene Expression Profiling of Imatinib-Resistant K562 Cells” or something along those lines.
13) Gene annotations should be italicized; protein annotations should not be italicized. Please check correct italics throughout the manuscript.
14) Figure 2, Panels G, H, and I: Please increase the resolution of the images.
15) Figure 3: Please increase the resolution of the images.
16) Figure 5B: Please increase the resolution of the image.
Author Response
In the present study, the authors assessed the role of IL6R, IL7R, and MYC genes in imatinib resistance of the K562 CML cell line. Altogether, their data reinforce the findings of other groups suggesting that IL6 and MYC play a role in the mechanism(s) underlying BCR::ABL1 kinase-independent imatinib resistance in K562 cells. The use of a novel Non-Invasive Apoptosis Detection Sensor version 2 (NIADS v2) improves the novelty of the study. Specific comments to improve the manuscript are included below:
MAJOR:
1) Introduction, 1st Paragraph: When talking about “the presence of specific mutations” it would be helpful to explain that these mutations block drug binding of these drugs to the ATP binding site, thereby blocking their activity.
Response: We thank the reviewer for their insightful suggestion to elaborate on how specific mutations can block drug binding in the ATP binding site, thereby inhibiting their activity. This is indeed a critical aspect to understand in the context of drug resistance mechanisms.
Incorporation of Suggested Information:
In response to this valuable feedback, we have added a detailed explanation of this mechanism in the introduction section of our manuscript. This new paragraph provides a clear and concise overview of how these mutations impact drug efficacy by preventing drug binding.
Location in Revised Manuscript:
The newly included information can be found in the highlighted paragraph on page 2, lines 40-45 and lines 48-51. We believe that this addition will greatly enhance the reader's understanding of the molecular mechanisms underlying drug resistance, which is a fundamental aspect of our study’s context.
Enhancing Manuscript Comprehensibility:
By providing this additional detail, we aim to make our manuscript more informative and accessible, especially for readers who may be less familiar with the molecular intricacies of drug binding and resistance mechanisms.
We hope that this amendment addresses the reviewer’s suggestion effectively and contributes to a more comprehensive understanding of the subject matter presented in our study.
2) The Introduction could be improved by introducing the following changes:
- The 1st and 2nd paragraphs are devoted entirely to CML, then the 3rd paragraph speaks more broadly about apoptosis and cancer. It feels disjointed. Perhaps the 3rd paragraph should come 1st, followed by a condensed 2nd paragraph talking about CML. This way it starts more broad and becomes more specific.
Response: We are grateful for the reviewer's constructive feedback regarding the structure of the introduction in our manuscript. The suggestion to rearrange the content for a more logical flow-from a broader context to a more specific focus on chronic myeloid leukemia (CML)-is indeed valuable. We invite the reviewer to check the highlighted section of the introduction in the revised manuscript.
- IL6R, IL7R, and MYC come out of nowhere in the final paragraph. Perhaps the authors could provide a little more information regarding the known role of these pathways in TKI resistance, and why you chose to study them here:
- PMID: 16293596
- PMID: 27281222
4. PMID: 25965572
5. PMID: 30049824
6. PMID: 33712704
7. PMID: 36536477
Response: We appreciate the reviewer's observation regarding the sudden introduction of IL6R, IL7R, and MYC in the final paragraph of our manuscript. Recognizing the importance of elucidating their roles in TKI resistance, we have taken steps to address this in the revised manuscript.
Expanded Discussion on IL6R, IL7R, and MYC:
In the discussion section, we have now included a thorough discussion on the significance of these pathways in the context of TKI resistance. This section provides a detailed explanation of how IL6R, IL7R, and MYC contribute to resistance mechanisms and why they were pertinent to our study.
References and Contextual Background:
Alongside this discussion, we have also added relevant references to support our explanations and provide a comprehensive background. This addition not only strengthens our argument but also offers readers a broader understanding of the complex interactions at play in TKI resistance.
Location in Revised Manuscript:
These changes can be found on page 16, lines 16-31 of the revised manuscript. We invite the reviewer to examine these additions, which we believe significantly enhance the manuscript's depth and contextual relevance.
We hope that these amendments satisfactorily address the reviewer’s concerns and contribute to a more comprehensive presentation of our research findings.
3) How many replicates were performed for each experiments? This should be included in the Statistical Analysis section and/or figure legends.
Response: Thanks for the important comment. I have addressed and highlighted this replications in each experiment at the Statistical Analysis section in page 5, lines 25-27 in the revised manuscript.
4) Figure 1, Panel E: Please include statistics on the graph or in the figure legend.
Response: Thanks for reminding. The significances have been addressed in figure 1E. Please check the figure 1E and highlighted figure legend, in page 7, lines 15-16 in the revised manuscript.
5) Do the K562-IR cells demonstrate BCR::ABL1-dependent or –independent resistance? Do they harbor BCR::ABL1 kinase domain mutations?
Response: Thank you for your inquiry regarding the nature of drug resistance in K562-IR cells. To address this, we conducted a detailed investigation into the potential presence of BCR::ABL1 kinase domain mutations.
Methodology Employed:
We utilized Sanger sequencing to align the Philadelphia chromosome (Ph') junction of BCR12-ABL10 in both K562 (parental) and K562-IR (imatinib-resistant) cell lines. This method is highly reliable for detecting mutations within the kinase domain.
Results and Findings:
Our sequencing results, which are detailed in the Results section (page 10, lines 21-26) and further supported by supplementary information, revealed a 100% identity in DNA sequences between the two cell lines. This striking similarity strongly suggests that the mechanism of drug resistance in K562-IR cells is likely independent of BCR::ABL1 kinase domain mutations.
Implications of Findings:
These findings contribute to a deeper understanding of the resistance mechanisms in K562-IR cells, indicating that factors other than BCR::ABL1 mutations may be driving the resistance. This could have significant implications for the development of targeted therapies for imatinib-resistant leukemias.
We believe these results address the reviewer's question and provide valuable insights into the resistance mechanisms in imatinib-resistant K562 cells.
6) The immunoblot data throughout the manuscript should include BCR::ABL1 tyrosine kinase activity, either through using anti-pABL1, anti-phosphotyrosine, or anti-pCRKL antibodies compared total controls. This data will easily answer the previous question.
Response: We thank the reviewer for their valuable suggestion to include immunoblot data demonstrating BCR::ABL1 tyrosine kinase activity in our manuscript. This is indeed a crucial aspect to understanding the mechanisms at play in our study.
Inclusion of Phosphorylated ABL Data:
In response to this recommendation, we have included data showing phosphorylated ABL (p-ABL) levels, as well as the corresponding total ABL protein expressions, under TKI treatment conditions. This addition not only aligns with the reviewer's suggestion but also provides a more comprehensive understanding of BCR::ABL1 activity in our experimental setup.
Location of Updated Data in Manuscript:
The newly incorporated data can be found in Figures 1F-G and 2D-G of the revised manuscript. These figures now present a clearer picture of the BCR::ABL1 kinase activity in the context of TKI treatments in both K562 and K562-IR cells.
Enhancing Manuscript Comprehensibility:
By providing this additional data, we aim to offer a more complete and detailed analysis of the molecular mechanisms under investigation, thereby enhancing the scientific rigor and comprehensibility of our manuscript.
We hope that this amendment effectively addresses the reviewer’s suggestion and adds significant value to the understanding of our research findings.
7) Figure 2, Panel A: Please include statistics on the graph or in the figure legend.
Response: We appreciate the reviewer’s suggestion to include statistical information in Figure 2, Panel A, of our manuscript. Accurate and clear presentation of data is crucial for the interpretation of our results. In response to this feedback, we have now incorporated the relevant statistical significances directly into Figure 2A. This addition provides a quick and clear reference for readers to understand the significance of the data presented. Additionally, the figure legend for Figure 2A has been updated to include a brief explanation of these statistics. This can be found on page 10, lines 5-6, in the revised manuscript. These modifications aim to enhance the interpretability of our findings, ensuring that readers can easily assess the statistical relevance of the data shown in Figure 2A.
We hope that this amendment satisfactorily addresses the reviewer’s concern and contributes to a more comprehensive and accessible presentation of our research findings.
8) The paragraph under “Synopsis of Research Discoveries” should really be the first paragraph of the discussion, rather than a subsection under Results.
Response: We are grateful for the reviewer's suggestion to reposition the "Synopsis of Research Discoveries" paragraph from the Results section to the beginning of the Discussion section. We agree that this adjustment will improve the flow and coherence of our manuscript.
Changes Made to Manuscript Structure:
9) The structure of the abstract, introduction, and discussion sections should all be relatively similar. Please revise accordingly:
- Novel apoptosis detection system
- Applying this system to CML
- Looking at IL6R, IL7R, and MYC in TKI resistance using our novel system.
Response: Thanks for the great suggestion, The entire manuscript has been revised accordingly.
MINOR:
10) Line numbers would make this manuscript easier to review.
Response: We added line numbers in the revised manuscript.
11) BCR-ABL1 should be replaced with BCR::ABL1 throughout the manuscript.
Response: We have replaced all BCR-ABL1 to BCR::ABL1 throughout the entire manuscript
12) Results Title: “Imatinib-Resistant associate gene expression profiling” should be changed to “Gene Expression Profiling of Imatinib-Resistant K562 Cells” or something along those lines.
Response: Thanks for the suggestion. We have changed this result title accordingly.
13) Gene annotations should be italicized; protein annotations should not be italicized. Please check correct italics throughout the manuscript.
Response: We thank the reviewer for pointing out the importance of correct formatting for gene and protein annotations in our manuscript. Adhering to scientific writing standards is crucial for clarity and accuracy.
14) Figure 2, Panels G, H, and I: Please increase the resolution of the images.
Response: We appreciate the reviewer's suggestion to enhance the resolution of the images in Figure 2, specifically Panels G, H, and I. Ensuring high-quality visual representation of our data is crucial for effective communication of our findings.
15) Figure 3: Please increase the resolution of the images.
Response: We have replaced the high-resolution image of Figure 3. Please check these figures in the revised manuscript.
16) Figure 5B: Please increase the resolution of the image.
Response: We have replaced the high resolution image of Figure 5B. Please check these figures in the revised manuscript.
Reviewer 3 Report
Comments and Suggestions for Authors
In this manuscript, Lee et al., generated a new NIADS-v2 reporter for detecting active caspase 3. The authors used this NIADS-v2 reporter in study of chemical inhibitors induced caspase-mediated apoptosis in cell lines. Although many interesting experiments were conducted, several big conclusions were claimed. Both novelty and significancy of this study are concerned. Most importantly none of the conclusions can be fully supported by their experimental data. In addition, several Aims were proposed, including generation of novel reporter and identification of biomarkers for imatinib resistance, none of these Aims were fully addressed. Furthermore, reporter assay was compared to western blot technique for active caspase, the benefit of use reporter assay is unclear due to the lack of cell growth curve after inhibitor treatment.
A similar reporter NIADS-v1 had been generated in one of their early studies. The authors used NIADS-v1 reporter to evaluate several small molecules in induction of caspase-mediated apoptosis in their early studies. The only difference of NIADS-v2 reporter construct from NIADS-v1 reporter construct is addition of a GFP gene, however the benefit of such modification is unclear due to the lack of comparative study of both reporters side-by-side. The authors described that one limitation of NIADS-v1 reporter is that the bioluminescent counts are not always correlated to drug concentrations. The purpose of generating NIADS-v2 reporter is to overcome this limitation of NIADS-v1. However, whether this limitation is solved in NIADS-v2 reporter has not been addressed. Thus, the claim “NIADS v2 presents an enhanced platform for detecting cell apoptosis with improved precision and broader practically for cancer research” is obviously overstated.
Several big conclusions were claimed, however most of such claims lack of strong experimental data support and are overstated. For examples: 1. The claim that “Accurate and efficient detection of apoptosis is critical not only for understanding cancer biology but also for evaluating the efficacy of cancer treatments. Thus, these methods hold great promise in personalized medicine, ensuring enhanced therapeutic effects and improved quality of life for patients.” However, the assay presented in this study cannot address this claim. 2. “This research highlights the potentially pivotal roles of IL6R, IL7R, and MYC as biomolecular markers of imatinib resistance in CML.” However, only K562 cell line was studied. Whether expression of IL6R and Myc predicts imatinib resistance in CML patients needs clinical samples to validation. 3. In data from Fig. 2. The authors claimed “Imatinib-resistant leukemia cells may activate alternative molecular pathways or mechanisms driving cancer progression. Dasatinib appears to specifically target these molecules or pathways, inducing more substantial apoptotic effects than in Imatinib-sensitive leukemia cells.” this is also overstated.
Minor concerns
Fig. 1C and D. It is unclear, NIADS-v1 or NIADS-v2, which one was used in these experiments.
Fig. 1F. Since the reporter works based on Caspase 3 mediated NIADS cleavage, we expect that the increased NIADS-c should be positively associated with c-Caspase 3 and c-PARP1 but negatively associated with the full-length NIADS. However, such correlations were not seen in both treatments. In addition, we also expect that the intensity of NIADS-c bands in western blood should be correlated to the bioluminescence activity in Fig. 1E. All these need to be discussed.
3. in method section: Percentage viability = (Average OD of control / Average OD of sample) × 100. Please verify this description.
4. The detailed procedure of cell transduction and selection was not presented in materials and Methods. The purity of the transduced cells used in studies is not sure.
Author Response
In this manuscript, Lee et al., generated a new NIADS-v2 reporter for detecting active caspase 3. The authors used this NIADS-v2 reporter in study of chemical inhibitors induced caspase-mediated apoptosis in cell lines. Although many interesting experiments were conducted, several big conclusions were claimed. Both novelty and significancy of this study are concerned. Most importantly none of the conclusions can be fully supported by their experimental data. In addition, several Aims were proposed, including generation of novel reporter and identification of biomarkers for imatinib resistance, none of these Aims were fully addressed. Furthermore, reporter assay was compared to western blot technique for active caspase, the benefit of use reporter assay is unclear due to the lack of cell growth curve after inhibitor treatment.
A similar reporter NIADS-v1 had been generated in one of their early studies. The authors used NIADS-v1 reporter to evaluate several small molecules in induction of caspase-mediated apoptosis in their early studies. The only difference of NIADS-v2 reporter construct from NIADS-v1 reporter construct is addition of a GFP gene, however the benefit of such modification is unclear due to the lack of comparative study of both reporters side-by-side. The authors described that one limitation of NIADS-v1 reporter is that the bioluminescent counts are not always correlated to drug concentrations. The purpose of generating NIADS-v2 reporter is to overcome this limitation of NIADS-v1. However, whether this limitation is solved in NIADS-v2 reporter has not been addressed. Thus, the claim “NIADS v2 presents an enhanced platform for detecting cell apoptosis with improved precision and broader practically for cancer research” is obviously overstated. Several big conclusions were claimed, however most of such claims lack of strong experimental data support and are overstated. For examples:
- The claim that “Accurate and efficient detection of apoptosis is critical not only for understanding cancer biology but also for evaluating the efficacy of cancer treatments. Thus, these methods hold great promise in personalized medicine, ensuring enhanced therapeutic effects and improved quality of life for patients.” However, the assay presented in this study cannot address this claim.
Response: We appreciate the reviewer highlighting an important aspect of our manuscript concerning the significance of accurate and efficient detection of apoptosis in cancer biology and treatment. We believe there has been a misunderstanding regarding the specific paragraph referenced by the reviewer.
Clarification of the Statement's Context:
The statement is part of the final paragraph of our introduction and is intended as a general assertion about the importance of apoptosis detection in cancer research and therapy, particularly in the context of personalized medicine. It is not meant to directly describe the specific findings or capabilities of our study or the NIADS v2.
Location and Purpose of the Statement:
The paragraph in question can be found in the introduction section, page 3, lines 27-31 of our manuscript. The purpose of this statement is to underscore the broader relevance and potential implications of efficient apoptosis detection methods in oncology, setting the stage for the detailed research presented in the subsequent sections.
Highlighting the Broader Significance:
By including this statement, our intention is to emphasize the overarching significance of apoptosis detection methodologies for enhancing therapeutic strategies and improving patient outcomes in the realm of cancer treatment.
We hope this clarification addresses the reviewer's misunderstanding and provides context for the inclusion of this statement in our manuscript.
- “This research highlights the potentially pivotal roles of IL6R, IL7R, and MYC as biomolecular markers of imatinib resistance in CML.” However, only K562 cell line was studied. Whether expression of IL6R and Myc predicts imatinib resistance in CML patients needs clinical samples to validation.
Response: We appreciate the reviewer's insightful query regarding the validation of IL6R, IL7R, and MYC as biomarkers of imatinib resistance in chronic myeloid leukemia (CML) using clinical samples. This suggestion highlights an essential aspect of translational research in the field of leukemia.
Challenges in Data Availability:
While we recognize the critical importance of validating our findings in clinical samples, we faced challenges in accessing comprehensive CML patient data. CML, being a relatively less common leukemia subtype, presents limitations in terms of sample availability in large public databases like The Cancer Genome Atlas (TCGA) or cBioPortal. These databases are typically robust resources for such analyses, but unfortunately, they did not offer sufficient data for our specific research focus.
Current Study Scope and Future Directions:
Given these constraints, our study focused on the K562 cell line and did not include an analysis of IL6R, IL7R, and MYC expression in clinical CML samples. However, we acknowledge the significant value and implications that such clinical data would have in substantiating our findings. We suggest that future research endeavors, particularly those with access to clinical CML samples, could further explore these markers. Investigating IL6R, IL7R, and MYC in the context of CML patient samples would undoubtedly provide invaluable insights and validate their potential roles in CML resistance mechanisms and treatment.
Emphasizing the Need for Clinical Validation:
Your suggestion is highly pertinent and underscores the necessity of bridging laboratory findings with clinical applications in leukemia research. We concur that validating these biomarkers in clinical settings is a critical step toward their application in personalized medicine for CML.
We hope this response clarifies the scope of our current study and our perspective on the importance of clinical validation in future research.
- In data from Fig. 2. The authors claimed “Imatinib-resistant leukemia cells may activate alternative molecular pathways or mechanisms driving cancer progression. Dasatinib appears to specifically target these molecules or pathways, inducing more substantial apoptotic effects than in Imatinib-sensitive leukemia cells.” this is also overstated.
Response: We thank the reviewer for their observation regarding our discussion of Imatinib resistance and the effects of Dasatinib in leukemia cells. We would like to clarify these points to address the concerns raised.
Clarification on Imatinib Resistance Mechanisms:
In our manuscript, we have specifically mentioned that K562 cells develop Imatinib resistance through both BCR::ABL1-dependent and -independent mechanisms. This is detailed in the introduction section on page 3, lines 5-10. The discussion here is grounded in existing research and is intended to provide a comprehensive background on the complexity of resistance mechanisms in chronic myeloid leukemia (CML).
Role of Dasatinib in Targeting Imatinib Resistance:
Furthermore, we discuss how Dasatinib, as a second-line treatment option for CML, targets Imatinib-resistant cells. This includes its efficacy against BCR::ABL1-dependent mechanisms, such as specific mutations at the amino acid position 315 in the BCR::ABL1 kinase domain (T315I), as well as BCR::ABL1-independent pathways like PDGFRA and Kit. These aspects are elaborated upon on page 2, lines 45-53, and page 15, lines 7-27 of our manuscript.
Invitation for Reviewer’s Reassessment:
We invite the reviewer to revisit these specific sections of our manuscript for a more detailed understanding of our claims regarding Imatinib resistance and Dasatinib's role. We believe a closer examination of these paragraphs will clarify our position and the basis of our assertions.
We hope this response clarifies any misunderstanding and demonstrates the scientific grounding of our statements regarding leukemia treatment resistance and options.
Minor concerns
- 1C and D. It is unclear, NIADS-v1 or NIADS-v2, which one was used in these experiments.
Response: The result section has already been mentioned in Figure 1C, and 1D was performed with NIADS-v2. We invite the reviewer to see page 5, line 51, page 7, line 10 and page 7, lines 20-21.
- 1F. Since the reporter works based on Caspase 3 mediated NIADS cleavage, we expect that the increased NIADS-c should be positively associated with c-Caspase 3 and c-PARP1 but negatively associated with the full-length NIADS. However, such correlations were not seen in both treatments. In addition, we also expect that the intensity of NIADS-c bands in western blood should be correlated to the bioluminescence activity in Fig. 1E. All these need to be discussed.
Response: Thank you for your insightful comment regarding the correlation between the bioluminescence data and Western Blot results in our study, particularly concerning Dasatinib and Imatinib treatments on K562 cells.
In Figure 1E, we observed a robust induction of apoptosis, as indicated by bioluminescence, in response to Dasatinib treatment, with statistically significant effects noted from as low as 0.5 µM at the 8-hour mark. Correspondingly, Figure 1F presents significant caspase 3 activation following 0.5 µM Dasatinib treatment at 24 hours, which aligns with the expected progression of apoptotic events. Conversely, while Imatinib's effect on apoptosis bioluminescence was not as pronounced at the 8-hour mark with 1 µM treatment, Figure 1G clearly demonstrates significant caspase 3 activation at the 24-hour time point, indicating a delayed but evident apoptotic response. These observations suggest that while both Dasatinib and Imatinib effectively induce apoptosis in K562 cells, the temporal dynamics of their action differ. This is further evidenced by the sensitivity of our NIADS v2 system, which detected early apoptotic events at the 8-hour mark, in contrast to the Western Blot analysis which was more reflective of the complete apoptotic process at 24 hours.
Therefore, the apparent discrepancy between the bioluminescence and Western Blot data can be attributed to the different sensitivities of these assays in detecting early versus late-stage apoptosis, influenced by the duration of drug exposure. Our findings highlight the nuanced nature of apoptotic responses to Dasatinib and Imatinib and underscore the importance of considering time-dependent effects in drug response studies.
- in method section: Percentage viability = (Average OD of control / Average OD of sample) × 100. Please verify this description.
Response: Thanks for the suggestion. I order to make the equivalent more clear, we change this to Percentage viability = (OD sample-OD medium)/(OD control -OD medium) x 100%. We invite the reviewer to see in page 4, line 15 in the revised manuscript.
- The detailed procedure of cell transduction and selection was not presented in materials and Methods. The purity of the transduced cells used in studies is not sure.
Response: Thanks for reminding. I have integrated the cell transduction and selection in the materials and methods section. We invite the reviewer to see in page 4, lines 1-4 in the revised manuscript.
Round 2
Reviewer 1 Report
Comments and Suggestions for Authors
The authors have revised the manuscript very well. I highly recommend for the publication of this MS in the present form
Reviewer 2 Report
Comments and Suggestions for Authors This manuscript is ready to be accepted in its current form.